# Structural insights into light-driven anion pumping in cyanobacteria

R. Astashkin[1,11], K. Kovalev [2,11], S. Bukhdruker[3,4,5,11], S. Vaganova[4,5], A. Kuzmin [6], A. Alekseev[6], T. Balandin[4,5], D. Zabelskii [7], I. Gushchin [6], A. Royant[1,3], D. Volkov [4,5], G. Bourenkov [2], E. Koonin [8], M. Engelhard [9], E. Bamberg[10] & V. Gordeliy[1,4,5] ✉

Transmembrane ion transport is a key process in living cells. Active transport of ions is carried out by various ion transporters including microbial rhodopsins (MRs). MRs perform diverse functions such as active and passive ion transport, photo-sensing, and others. In particular, MRs can pump various monovalent ions like $Na^+$, $K^+$, $Cl^-$, $I^-$, $NO_3^-$. The only characterized MR proposed to pump sulfate in addition to halides belongs to the cyanobacterium *Synechocystis* sp. PCC 7509 and is named *Synechocystis* halorhodopsin (*Sy*HR). The structural study of *Sy*HR may help to understand what makes an MR pump divalent ions. Here we present the crystal structure of *Sy*HR in the ground state, the structure of its sulfate-bound form as well as two photoreaction intermediates, the K and O states. These data reveal the molecular origin of the unique properties of the protein (exceptionally strong chloride binding and proposed pumping of divalent anions) and sheds light on the mechanism of anion release and uptake in cyanobacterial halorhodopsins. The unique properties of *Sy*HR highlight its potential as an optogenetics tool and may help engineer different types of anion pumps with applications in optogenetics.

Microbial rhodopsins (MRs) constitute a large family of photoactive 7 transmembrane (TM) α-helix proteins containing a retinal cofactor that is covalently bound to a conserved lysine residue in the 7th TM helix via a retinal Schiff base (RSB). While in the dark (or the ground) state of a rhodopsin, retinal is usually in the all-trans configuration, but upon light absorption, it isomerizes to the 13-cis conformation and triggers a sequence of changes in the structure as well as in absorption spectrum in the protein, collectively known as the photocycle. During the photocycle, the rhodopsin passes through several metastable intermediate states. Transitions between these states result in structural rearrangements of the protein and thus determine its function. In

nature, MRs perform diverse functions, such as ion pumping and channeling as well as sensory activity and even control of the enzymatic activity[1,2]. In particular, some MRs actively transport different monovalent ions like $H^+$, $Na^+$, $K^+$, $Cl^-$, $I^-$, $NO_3^-$.

The first microbial rhodopsin able to pump anions (specifically, chloride) was identified in the archaeon *Halobacterium salinarum* in 1977 and later named *Halobacterium salinarum* halorhodopsin (*Hs*HR)[3,4]. In 1986, another halorhodopsin (HR) was discovered in the extremely haloalkaliphilic archaeon *Natronomonas pharaonis* and was named *Np*HR[5]. It was shown that these two rhodopsins are organized in trimers and function as inward $Cl^-$ pumps[6]. Although metagenomics

[1]Univ. Grenoble Alpes, CEA, CNRS, Institut de Biologie Structurale (IBS), Grenoble, France. [2]European Molecular Biology Laboratory, Hamburg unit c/o DESY, Hamburg, Germany. [3]European Synchrotron Radiation Facility Grenoble, Grenoble, France. [4]Institute of Biological Information Processing (IBI-7: Structural Biochemistry), Forschungszentrum Jülich, Jülich, Germany. [5]JuStruct: Jülich Center for Structural Biology, Forschungszentrum Jülich, Jülich, Germany. [6]Research Center for Molecular Mechanisms of Aging and Age-related Diseases, Moscow Institute of Physics and Technology, Dolgoprudny, Russia. [7]European XFEL GmbH, Schenefeld, Germany. [8]National Center for Biotechnology Information, National Library of Medicine, National Institutes of Health, Bethesda, MD, USA. [9]Department Structural Biochemistry, Max Planck Institute of Molecular Physiology, 44227 Dortmund, Germany. [10]Max Planck Institute of Biophysics, Frankfurt am Main, Germany. [11]These authors contributed equally: R. Astashkin, K. Kovalev, S. Bukhdruker. ✉e-mail: valentin.gordeliy@ibs.fr

studies led to the discovery of numerous additional archaeal HRs, functional and structural studies were carried out almost exclusively on HsHR and NpHR[7]. A distinct feature of archaeal HRs is the 3-letter motif TSA instead of the DTD motif (D85, T89, and D96 in the 3th TM helix) that is characteristic of light-driven proton pump bacteriorhodopsin (BR)[8]. In BR, D85 plays a role of primary proton acceptor from the RSB, while T89 likely mediates this initial proton transfer and stabilizes the deprotonated form of the RSB in the course of photocycle[9]. D96 plays a role of proton donor to the RSB and is located at the cytoplasmic side of BR[10]. The HRs bind chloride near the RSB in the ground state and have a specific photocycle, which lacks the M intermediate state[7]. NpHR and its modification eNpHR3.0 are among the major optogenetic tools[11].

In 2014, chloride-pumping rhodopsins of a new type were identified in the flavobacterium *Nonlabens marinus*[12]. These rhodopsins differ from archaeal chloride pumps and in particular contain an NTQ motif, show different organization in the RSB region, and probably form pentameric assemblies in cell membranes, in analogy to related NDQ rhodopsins[13,14].

Another clade of chloride-pumping rhodopsins was discovered in 2015 in cyanobacteria. These proteins contain predominantly the TSD motif with TSV/TSL/TSI variations[15,16]. To date, only two rhodopsins from this clade have been experimentally characterized, namely MrHR (or MastR) from *Mastigocladopsis repens* and SyHR from *Synechocystis* sp. PCC 7509[17]. These proteins share 68% amino acid sequence identity and both contain the TSD motif. Both MrHR and SyHR are believed to be inward chloride pumps; however, measurements of pH changes in suspensions of *E. coli* cells that express SyHR have suggested that this protein may also pump sulfate in addition to halides. Thus, SyHR is currently the only known natural MR that is proposed to pump divalent anions. Notably, MrHR can also be turned into a potential sulfate pump through the introduction of SyHR-mimicking mutations what demonstrates their close similarity[18].

In general, cyanobacterial and archaeal HRs have some common features, such as chloride ion binding to the RSB in the ground state, trimeric organization in lipid membranes, and the absence of the M state in their photocycle. However, there are also notable differences between the proteins from these two different clades. As indicated above, archaeal HRs that have been experimentally studied so far cannot pump divalent anions. Whereas MrHR can be converted by a single mutation (T74D) into a proton pump[15], even ten BR-mimicking mutations cannot turn NpHR into a proton pump[19].

Another distinctive feature of cyanobacterial HRs is the presence of aspartate (D85 in SyHR) at the position corresponding to that of primary proton donor D96 in BR. This aspartate is substituted by alanine in archaeal HRs. Like in BR, this amino acid is deprotonated during the photocycle. However, the exact role of this aspartate in the anion transport unclear. It has been hypothesized that this residue is essential to prevent the backflow of chloride during ion transport against the high electrochemical gradient[16].

To date, all microbial rhodopsins that are considered to pump anions belong to one of these three families: archaeal HRs, NTQ bacterial chloride pumps, and cyanobacterial anion pumps (Fig. 1A)[20].

Currently, most of the structural studies were performed with light-driven anion pumps from archaea, such as HsHR and NpHR. In total, 16 3D structures of HRs have been reported including anion-free, chloride-, bromide-, azide-, nitrate-bound structures, and also structures of the L[1], N, O, and M-like intermediate states[21–23].

Structural data on bacterial Cl⁻ pumps are represented by 45 different structures of one protein, rhodopsin 3 from *Nonlabens marinus* (NM-R3, or NmClR). This set includes the structures of the wild type of NM-R3 at different temperatures and pH values, as well as its functional mutants and different active states structures obtained using time-resolved crystallography[24–26].

By contrast, structural information on cyanobacterial anion pumps is limited. Only in 2020 two research groups independently published structures of MrHR[15,27]. Additionally, authors of one of these works presented the structure of functional mutants of MrHR, including its $SO_4^{2-}$-pumping variant[18]. These data provided initial information on the anion-pumping mechanisms of cyanobacterial HRs.

Here we present the high-resolution structures of the native light-driven anion pump SyHR. We obtained four high-resolution structures of SyHR: Cl⁻ and $SO_4^{2-}$-bound forms of the ground state and the K and the O intermediate states of the Cl⁻ - pumping mode of the SyHR photocycle. The data sheds light on the chloride pumping mechanism,

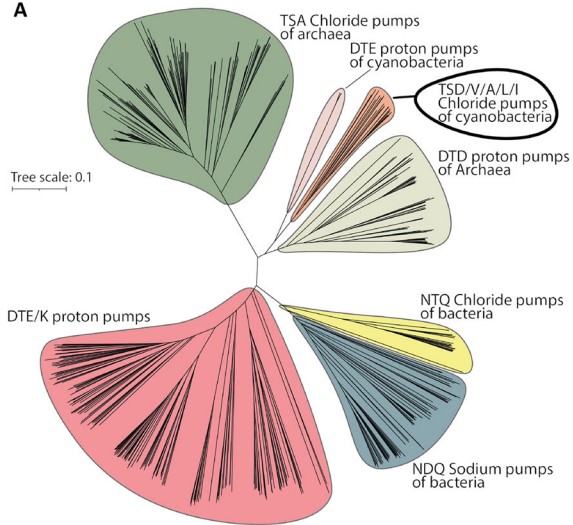
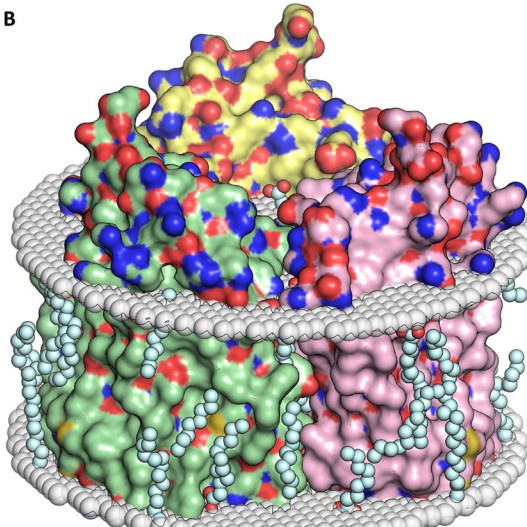

**Fig. 1 | Phylogenetic tree of microbial rhodopsins and organization of SyHR trimer in the membrane. A** Phylogenetic tree of prokaryotic microbial rhodopsins acting as light-driven ion transporters. Conducted ions and alternating three letter motifs (85, 89 and 96 amino acid positions according to HsBR) are specified for each clade. Tree scale is indicated by a horizontal line (substitutions per site). **B** Organization of trimer of ground state of chloride bound SyHR in membrane.

Protein molecules are shown with surface representation. Three protomers within the trimer are colored yellow, green, and rose, respectively. Hydrophobic/hydrophilic membrane core boundaries were calculated using PPM server and are shown with gray surfaces. Ordered lipid fragments observed in the electron densities are colored cyan. The cytoplasmic side is at the top of the figure.

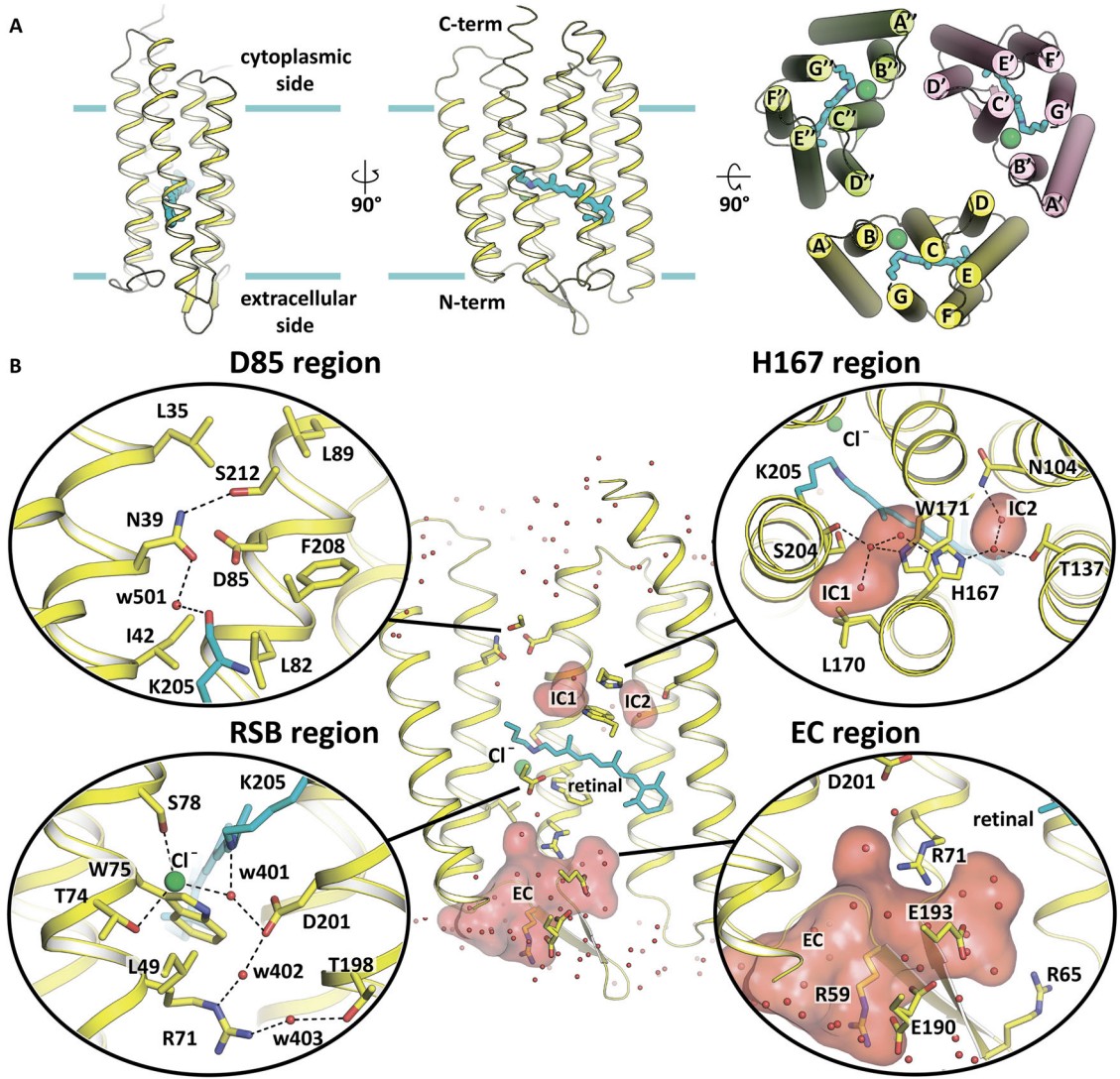

**Fig. 2 | Structure of SyHR protomer. A** Overall view of the SyHR protomer in the membrane. Calculated hydrophobic-hydrophilic membrane boundaries are shown as blue lines. Chloride ion is shown as a green sphere. The right section shows the view of the SyHR trimer from the cytoplasmic side. **B** Detailed view of the SyHR key compartments, including D85, H167, RSB, and EC regions. The cofactor retinal and K205 are colored cyan. Internal cavities were calculated using HOLLOW and are shown as red surface. Hydrogen bonds are indicated as black dashed lines. Water molecules are shown as red spheres.

which is quite similar to that of archaeal HRs. Particularly, the molecular basis for the high affinity chloride-binding site could be unravelled and that SyHR does not possess a sulfate-binding site within the protein.

## Results

### Crystallization of SyHR

We crystallized the protein using *in meso* approach similar to our previous works[28] using either chloride and sulfate containing buffers. Hexagonal crystals up to 100 µm in size were grown within 2 months. The obtained crystals were of two colors: red and violet, with maximum absorption wavelength of 536 nm and 558 nm, respectively, as determined by in crystallo UV–vis absorption spectroscopy[29]. Red crystals were grown in the precipitant solutions containing Cl⁻, while violet crystals were grown in the absence of Cl⁻ (Supplementary Fig. 1A, B). Crystals of both colors were of P 3 2 1 symmetry and contained one molecule of SyHR in the asymmetric unit. Analysis of the crystal packing indicated that SyHR is in its trimeric form in the crystals (Supplementary Fig. 1C).

### The overall structure of the chloride-bound SyHR in the ground state

Using red crystals of SyHR we solved the Cl⁻-bound structure of the protein at 1.57 Å resolution. The high-resolution structure reveals 126 protein-associated water molecules, including 27 inside SyHR, and 20 lipid fragments surrounding each protomer (Fig. 1B). Residues 1-225 from the total 234 of the SyHR molecule are resolved in the model. The SyHR protomer consists of 7 transmembrane α-helices (A-G) connected by three intracellular and three extracellular loops like in other type I rhodopsins. The BC extracellular loop is 13 residues long and contains a β-sheet formed by two β-strands (Fig. 2A).

As already mentioned, SyHR is organized into trimers in the crystals. Similar trimers are known for many MRs, including archaeal H⁺ and Cl⁻ pumps, but also MrHR from the same clade of cyanobacterial anion-pumping rhodopsins. Interprotomer contacts are formed via pairs of residues A46 - Y106 and Q57 - W126 of the helices B with D and BC loop with helix E respectively. In contrast to all previously solved structures of Cl⁻ pumps, we did not detect any Cl⁻ ions on the surface of SyHR.

## Retinal-binding pocket and the RSB region of *Sy*HR

A retinal chromophore is covalently bound to K205 and is in the all-*trans* configuration in the ground state of *Sy*HR. As in archaeal HRs, a Cl⁻ ion is found in the RSB region of *Sy*HR in its ground state (Supplementary Fig. 2A). The ion is coordinated by the T74 and S78 side chains of the characteristic TSD motif and also a water molecule (w401). The water molecule is further H-bonded to the RSB and D201. The distance between the RSB nitrogen and the Cl⁻ ion is 3.8 Å (Fig. 2B).

It should be noted that the retinal-binding pocket of *Sy*HR is almost identical to that of *Mr*HR. The similar retinal environment is in line with the similar maximum absorption wavelengths of the two proteins in their Cl⁻-bound state (536 and 537 nm for *Sy*HR and *Mr*HR, respectively).

## Organization of the ion release region of *Sy*HR

The structure of *Sy*HR in the ground state reveals two internal cavities at the cytoplasmic half of the protein, which we denoted as intracellular cavity 1 (IC1) and intracellular cavity 2 (IC2). The cavities are located in the region of H167 and are separated by the side chain of this residue. The presence of H167 is characteristic for only the clade of cyanobacterial anion-pumping rhodopsins. This histidine is absent in both archaeal HRs and eubacterial Cl⁻ pumps. It is located in helix F and it is stabilized in *Sy*HR by the H-bonds with two water molecules (Fig. 2B).

The size of IC1 in *Sy*HR is dictated by the conformation of the L170 side chain, which adopts two alternative orientations with the same occupancies of 0.5. In one of these, L170 points towards the lipid bilayer, which results in the enlargement of IC1. In this case, IC1 is filled with three water molecules (only this conformation is shown on the Fig. 2B). In the second conformation, the CD1 atom of the L170 side chain is moved closer to the inside of the *Sy*HR protomer. In this configuration, there are only two water molecules in IC1. Water molecules are H-bonded to each other and are stabilized by the side chains of W171 and H167 and also by the carbonyl oxygen of S204. Interestingly, the analogous cavity to IC1 is found also in *Mr*HR. However, in *Mr*HR, the cavity is smaller and is filled with only one water molecule, which does not interact with the characteristic histidine residue of the helix F (H166 in *Mr*HR).

IC2 is characteristic only for *Sy*HR, but not for *Mr*HR. This is due to the presence of polar T137 in *Sy*HR at the position of hydrophobic I136 in *Mr*HR. Consequently, the IC2 is a relatively compact hydrophilic cavity filled by two water molecules, coordinated by N104, T137, and H167 residues. In total, there are five water molecules in the region of H167 in *Sy*HR, compared to only one water molecule near H166 in *Mr*HR.

Another feature of the cyanobacterial anion-pumping rhodopsins is the presence of D85 in helix C at the cytoplasmic side of the protein. This residue belongs to the characteristic TSD motif. As mentioned above, it is functionally analogous to the proton donor D96 residue of BR. It should be noted that a corresponding aspartate is absent in the archaeal HRs as well as in the bacterial Cl⁻ pumps. This residue is replaced by alanine in archaeal HRs and by glutamine in bacterial Cl⁻ pumps. In *Sy*HR, D85 is stabilized by the H-bonds with N39. N39 is further stabilized by the H-bonds with the water molecule w501 and S212. Water w501 is also connected with the carbonyl oxygen of K205, to which the retinal is attached. The area is surrounded by a hydrophobic region formed by L35, L89, F208, I42, and L82 residues (Fig. 2B). A similar organization is found in *Mr*HR; the only minor difference is that in *Sy*HR the S212 side chain occupies two alternative conformations.

## The extracellular part of *Sy*HR

The extracellular (EC) half of the *Sy*HR protomer forms a large aqueous concave basin, protruding from the bulk to R71 (counterpart of R82 residue in BR) and, surprisingly, even further to the RSB counterion

D201. The basin is filled with 18 water molecules, which support a dense H-bond network in this region. The basin is surrounded by R71, E190, E193, R65, T194, Y72, W178, R59, Y50, L6, M53, T198, Y197, T67, T183, N182, and T115 (Fig. 2B). E190 adopts two different conformations. The presence of such a basin at the extracellular side is likely a common feature of cyanobacterial anion-pumping rhodopsins as it is also found in *Mr*HR; however, the shape and size of the basin vary between the models (PDB IDs: 6XL3 and 6K6I). The major difference is in the position of the side chain of Q189. The analysis of the electron density maps of the 6XL3 model shows that Q189 is flexible and might adopt two alternative conformations. In the 6K6I model, the side chain of Q189 is poorly resolved, with the average B factor of more than 80 Å² compared to the mean value 31 Å² for the protein atoms.

Yun et al.[18] designed $NO_3^-$- and $SO_4^{2-}$-pumping variants of *Mr*HR by introducing of the *Sy*HR-mimicking mutations to the extracellular region of the protein, which serves as an entrance for anions. Experiments showed evidences of $SO_4^{2-}$-pumping for several *Mr*HR mutants, such as E182A, N63A/P118A, and N63A/P118A/E182A. The structure of the N63A/P118A variant of *Mr*HR was also reported and showed enlarged concavity at the extracellular part of the protein. However, the basin was enlarged in the mutant in the region between BC-loop and helices D, E, and F, while in *Sy*HR the pore for anion uptake is located near helices A, B, and G.

## Structure of the $SO_4^{2-}$-bound form of *Sy*HR

As mentioned above, we obtained not only red but also violet crystals of *Sy*HR. The precipitant solution in this case contained 2.0 M of $SO_4^{2-}$ and lacked Cl⁻ ions. Moreover, the protein used for crystallization was purified without Cl⁻. These crystals allowed us to solve the structure of the protein at 2.0 Å resolution. Since the concentration of $SO_4^{2-}$ in precipitant solution was much higher than its $K_d$ value of *Sy*HR for this type of ions[17] (5.81 mM) and the maximum absorption wavelength of the protein in violet crystals is similar to that reported for the $SO_4^{2-}$-bound form of *Sy*HR in solution[17] (556 nm), we assume that the determined structure represents *Sy*HR in the $SO_4^{2-}$-bound form. However, a shoulder at 536 nm (Supplementary Fig. 3A) indicated the presence of chloride.

Since *Sy*HR has low $K_d$ to Cl⁻ (0.112 mM), even after using Cl⁻-free solutions during protein purification and crystallization, we found that approx. 25.5% of protein molecules within the violet crystal contain Cl⁻ ion bound near the RSB. This might be due to the trace amounts of Cl⁻ in the high-molarity salts used as precipitants for crystallization or co-purification of chloride ions with SyHR. Nevertheless, the major fraction (74.5%) is Cl⁻-free. This high concentration of Cl⁻-free molecules enabled to determine the structure of violet *Sy*HR (*Sy*HRᵛ) in the presence of sulfate.

*Sy*HRᵛ contains one $SO_4^{2-}$ ion on its surface. The ion was found at the cytoplasmic side of the protein (Fig. 3A). The ion is placed between the helices D and E. It interacts with the Q148 of the helix E and interacts with the S93 of the helix D through a water molecule. It is also within 3.2 Å from the K33 of the neighboring protomer. The sulfate ion is distant from the RSB (28 Å) (Fig. 3C). The positions of the $SO_4^{2-}$-coordinating residues are similar in the Cl⁻- and $SO_4^{2-}$-bound structures of *Sy*HR.

For further characterization of the bound $SO_4^{2-}$ ion, we conducted 500 ns long molecular dynamics (MD) simulations of *Sy*HRᵛ trimer in presence of sulfate ions (Supplementary Fig. 4A, B. Atomistic model of the starting structure used in simulations is available as Supplementary Data 1. Average densities of sulfate sulfur atoms obtained in the simulations are available as Supplementary Data 2). We observed multiple binding and unbinding events, which reflects the dynamic nature of these protein-ion interactions (Supplementary Fig. 5). The majority of binding events are observed in almost the same place of sulfate ion position in the crystallographic structure (Supplementary Fig. 4B). Slight variation in the position of the sulfate could be due to

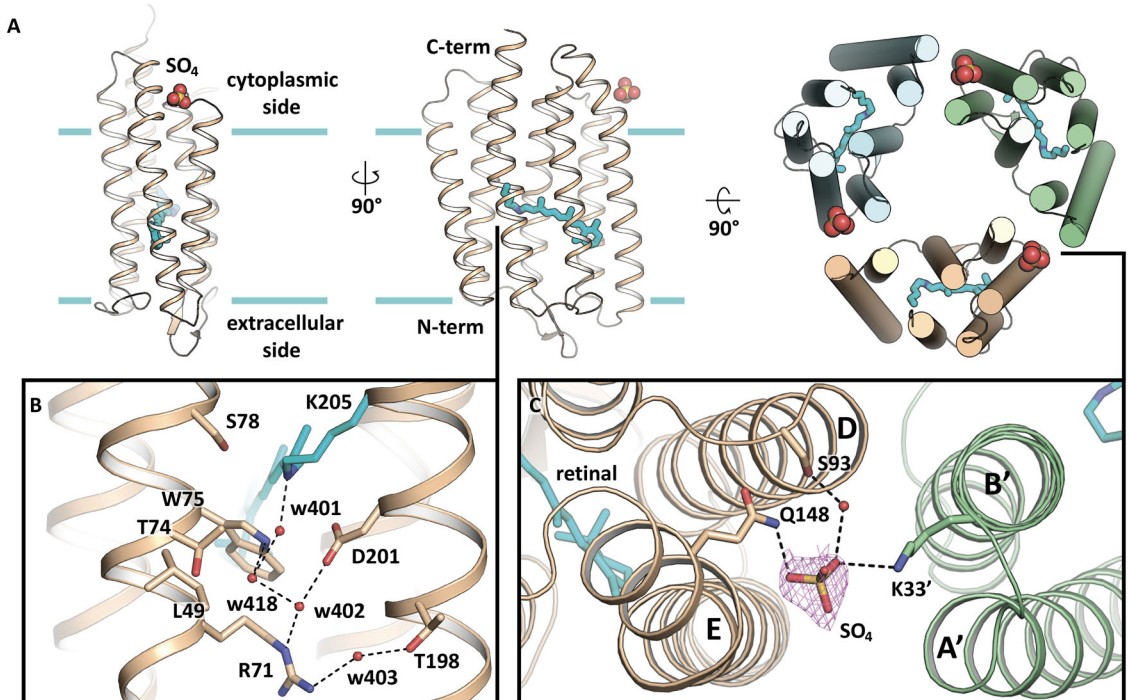

**Fig. 3 | Structure of SO₄²⁻-bound form of *Sy*HRᵛ. A** Overall view of the *Sy*HRᵛ protomer in the membrane. Calculated hydrophobic-hydrophilic membrane boundaries are shown as blue lines. Sulfate ion is shown with spheres. The right section represents the view of the *Sy*HRᵛ trimer from the cytoplasmic side. **B** Structure of the RSB region. **C** Coordination of sulfate ions between two protomers. *2Fo-Fc* composite omit map is countered around the sulfate ion at 1sigma level. The cofactor retinal and K205 are colored cyan.

the fact that the crystal structure was obtained at cryotemperature, while the simulation was done at 303.15 K.

We note that in addition to the SO₄²⁻ ion described above, we observed an ion-like density on the surface of the *Sy*HR trimer near the R146 side chain. This density was initially ascribed to a SO₄²⁻ ion in the crystallographic model. However, enrichment of ions at this site was not confirmed in the MD simulation. Besides the occasional contacts with sulfates, R146 side chain was observed to make contacts with the phosphate moieties of lipids (POPC). Consequently, we did not assign a sulfate ion to this density in the final crystallographic model. This example highlights the usability of MD simulations for interpretation of electron densities resulting from crystallographic experiments, complementing similar observations made for other proteins[30].

Surprisingly, our structural data showed that there was no SO₄²⁻ ion close to the RSB in *Sy*HRᵛ. This stands in contrast with the earlier spectroscopy study, where it was proposed that SO₄²⁻ occupies the same site as Cl⁻ near T74 and S78 residues[17]. The hypothesis was supported by the notable red shift (542 nm to 556 nm) of the absorption peak with the addition of SO₄²⁻ to the detergent-solubilized *Sy*HR in the absence of Cl⁻. By contrast, in the presence of Cl⁻, SO₄²⁻ does not affect the *Sy*HR spectrum. Thus, analysis of our structure, together with the spectroscopy of *Sy*HRᵛ in crystals and MD simulations leads to the conclusion that such red-shifted maximum absorption wavelength is observed without SO₄²⁻ binding near the RSB. We suggest that there are long-distance interactions of the surface-bound SO₄²⁻ ion with the RSB, resulting in the red shift of the spectrum of *Sy*HR. For instance, such interactions have been recently demonstrated for a light-driven Na⁺ pump KR2[31]. In KR2, similar to the SO₄²⁻-bound form of *Sy*HR, the transported substrate is not bound in the core of the protomer in the ground state. Instead, in both rhodopsins, ions are located on the protein surface, near the putative ion-release regions. Because no structure of *Sy*HR in the absence of both Cl⁻ and SO₄²⁻ is available, the exact influence of the surface-bound ion on this protein is not completely clear at the moment.

The RSB region is altered in the SO₄²⁻-bound form of *Sy*HR compared to that of the Cl⁻-bound form. In general, the organization of the RSB pocket is similar to that of the anion-free blue form of *Np*HR[21]. T74 is reoriented notably and moved closer to the RSB, while the side chain of L49 is flipped outside of the protomer to allow enough space for T74 (Fig. 3B). As a result, the positions of residues 71-78 of the helix C are also altered in *Sy*HRᵛ. Consequently, the side chain of R71 is moved towards the extracellular space by 1 Å. However, the latter has only a minor effect on the positions of the surrounding water molecules and residues in the extracellular basin.

Notably, the synchronous rearrangements of T74 and L49 (Fig. 3B) are similar to those observed in the light-driven sodium pump KR2. Indeed, the L74 and N112 residues of KR2, which are analogs of L49 and T74 of *Sy*HR, are reoriented in the same manner in course of photocycle upon Na⁺ binding in the core of the KR2 protomer[32].

## Cryotrapping and assignment of the intermediate states of the *Sy*HR photocycle in crystals

High-quality red crystals of *Sy*HR were used for the cryotrapping of the intermediate states of the Cl⁻-pumping mode. To this end, we used a modification of our previous approach[32]. With violet crystals, due to their worse diffraction quality, we were not successful in solving the structures of the active states.

We validated the cryotrapped intermediates using in crystallo UV–vis absorption spectroscopy on *Sy*HR crystals at the *ic*OS Lab of the European Synchrotron Radiation Facility (ESRF, Grenoble, France)[29].

The analysis of this structural data together with the previously reported data on *Sy*HR and other anion-pumping rhodopsins allowed us to provide a model of the observed structural rearrangements to the K and O intermediates of the Cl⁻-pumping mode of the *Sy*HR photocycle in solution.

First, we cryotrapped the late O intermediate state of the Cl⁻-pumping mode of the *Sy*HR photocycle. Because the O state is the

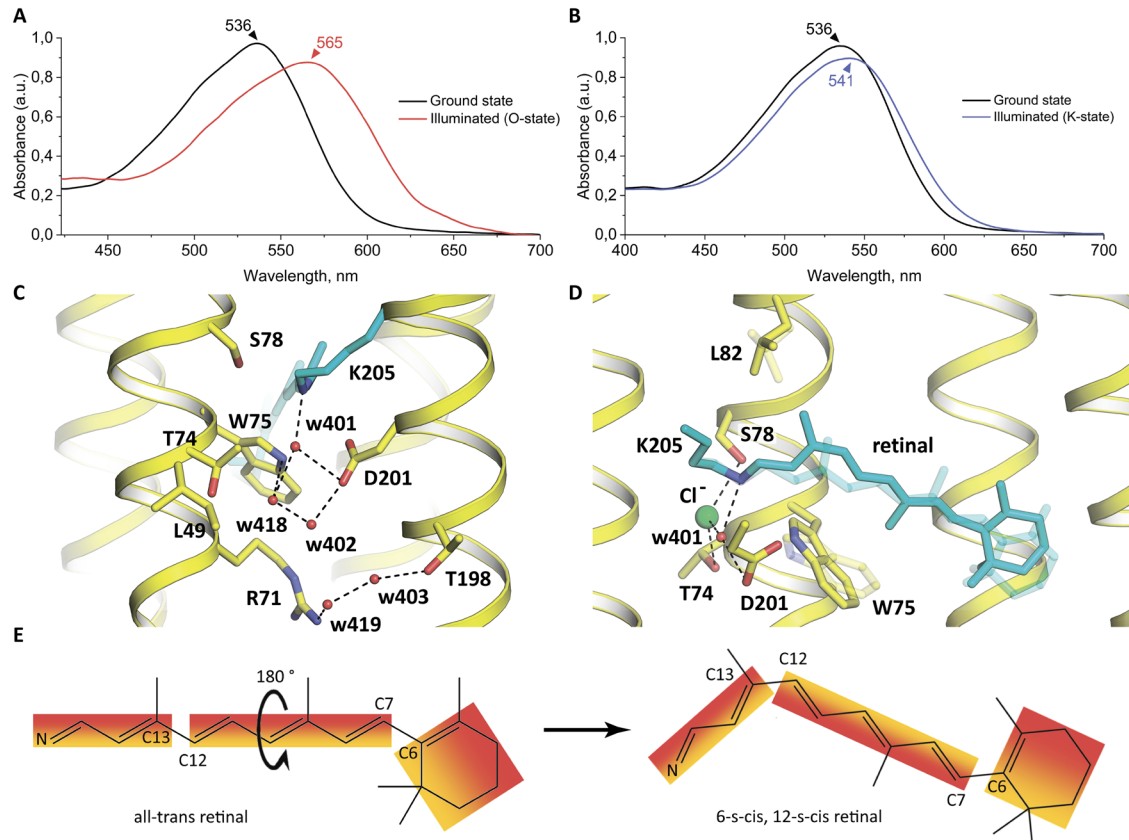

**Fig. 4 | Structures and spectra of active states. A** Cryotrapping of the O state of the *Sy*HR photocycle in crystals. Visible light absorption spectra of the red crystals of *Sy*HR before (black line) and after (red line) illumination by the 532-nm laser for 2 seconds at 293 K. **B** Cryotrapping of the K state of the *Sy*HR photocycle in crystals. Visible light absorption spectra of the red crystals of *Sy*HR before (black line) and after (blue line) illumination by the 532-nm laser for 1 minute at 100 K. **C** Structure of the RSB region in the O state. (**D**) Structure of the retinal-binding pocket in the K state. Ground state positions of amino acids and retinal are shown as transparent. The cofactor retinal and K205 are colored cyan. (**E**) The scheme of the retinal isomerization from all-*trans* to 6-s-cis, 12-s-cis form. The fixed parts of the retinal are shown in the form of colored blocks.

dominant intermediate of the photocycle, it could be accumulated upon continuous green light illumination at room temperature. For the accumulation and trapping of the O state, we applied the following procedure. When the crystal was in the cryostream, we blocked the cryostream for 2 s while illuminating the crystals with a 532-nm laser. UV–vis absorption spectra showed the appearance of the red-shifted state fraction corresponding to presence of the O state in the *Sy*HR crystals after the procedure (Fig. 4A). The following structural analysis indicated the absence of the Cl⁻ ions inside the protomer in the corresponding fraction, which is expected for the O state of *Sy*HR. Therefore, the accumulated state was assigned to the O state of the rhodopsin.

Second, we aimed at accumulation and cryotrapping of the K state of the *Sy*HR photocycle. It is known that in the K state the structural rearrangements in all studied MRs are small; typically, only the retinal and several surrounding residues are slightly affected. Such rearrangements may occur at cryogenic temperatures. For instance, in BR, the K state could be formed at 100 K[33]. In KR2, the K state was obtained at an even lower temperature of 77 K[34]. In our work we accumulated trapped K state by illuminating *Sy*HR crystals with a laser at 532 nm for 1 min at 100 K, resulting in appearance of a red-shifted intermediate (Fig. 4B). As it described below, our structural analysis showed that the rearrangements associated with the formation of the accumulated state only occur in the retinal cofactor and two adjacent residues. Thus, taking into account the low temperature of trapping, red-shift of the absorption maximum, and only local corresponding structural changes in the *Sy*HR molecule, we

tentatively assigned the accumulated state to the K state of the protein photocycle.

Importantly, in a test experiment, after the cryotrapping of both intermediates, the crystals were heated up to room temperature without laser illumination by blocking the cryostream for 2 s, for an additional validation of the reversibility of cryotrapping of *Sy*HR in crystals. Indeed, immediately after heating, the crystal contained a fraction of the protein in the cryotrapped intermediate, which rapidly relaxed to the ground state, as demonstrated by spectrophotometry (Supplementary Fig. 3B, C). Thus, the observed red shifts of the maximum absorption wavelengths in the cases of the cryotrapping of the K and O states are not associated with non-reversible changes in *Sy*HR, such as, for instance, protein crystals dehydration.

### Structure of the O state of the *Sy*HR photocycle

The O state of the *Sy*HR photocycle was solved at 1.6 Å-resolution. Crystallographic data analysis showed that the major fraction of the protein molecules in the crystal are in the O state. $2F_o$-$F_c$ and difference $F_{oO}$-$F_{oGr}$ electron density maps clearly show the structural rearrangements in *Sy*HR associated with the O state formation (Supplementary Fig. 2B, Supplementary Fig. 6A). It needs to be noted, that $F_o$-$F_c$ difference electron density maps showed a trace amount of the ground state, which was insufficient for proper fitting of this residual conformation (Supplementary Fig. 2D). Therefore, we fitted the crystallographic data exclusively with the O state structure in the final model.

Major conformational changes in the structure of the O state, in comparison to the ground state, are observed exclusively in the area

around the RSB (Fig. 4C). Interestingly, the O state structure of the RSB region is closely similar to the $SO_4^{2-}$-bound form in the ground state, which also lacks the chloride ion near the RSB. In particular, we observed the same flip of L49 and rearrangement of T74. The retinal is in the all-*trans* configuration in the O state (Supplementary Fig. 2C).

The only major difference between the O state and the $SO_4^{2-}$-bound form is the position of R71, which is slightly shifted towards the extracellular side in the O state structure. This change led to rearrangement of H-bonds and appearance of a new water molecule w419, not found in either the ground state or $SO_4^{2-}$-bound form. Thus, in the O state, R71 forms an H-bond with this water w419 instead of w403. In its turn, w419 is H-bonded to w403, which is coordinated by T198 (Fig. 4C).

### Structure of the K state

The cryotrapped K state of *Sy*HR was solved at 1.7 Å resolution. $2F_o$-$F_c$ electron density maps showed that the major fraction of the proteins in the crystal remains in the ground state of the Cl⁻-bound form of *Sy*HR, which is consistent with the results of the microspectrophotometry experiments (Fig. 4B). The difference $F_{oK}$-$F_{oGr}$ electron density maps indicated the presence of the strong signals above the level of 5σ near the C12, C13, C19, and C20 atoms of the retinal cofactor and also around its β-ionone ring as well as near the W75 and L82 side chains (Supplementary Fig. 6B). The analysis of the structural data indicated that the fraction of the K state in the crystal was nearly 35%. To build the K state model more accurately, we used extrapolated electron density maps. It should be noted that in the case of K state the datasets corresponding to the dark and illuminated states were collected from the same crystal of *Sy*HR, which resulted in high isomorphism and reliable difference electron density maps for building of the intermediate state structure.

The data showed structural rearrangements in the retinal region of *Sy*HR in the K state (Fig. 4D). Surprisingly, the retinal cofactor was found to be in an unusual 6-s-cis 12-s-cis configuration as unequivocally indicated by $F_{oK}$-$F_{oGr}$ difference electron density maps (Supplementary Fig. 6B). Despite the fact that such configuration of retinal is peculiar and has never been observed in microbial rhodopsins before, it is the only one that could be fitted into the electron density maps. This configuration means that the RSB and β-ionone ring retained their orientation, while the polyene chain rotated 180 degrees from atoms C7 to C12. In general, the following changes have occurred in the retinal (Fig. 4E). The C20 atom of the retinal is thus shifted by 1.7 Å towards the L82 residue, which itself is reoriented to allow enough space for the cofactor isomerization. At the same time, the C19 atom of the retinal is shifted closer to the W75 residue. Consequently, the side chain of W75 is rotated by 10 degrees (Fig. 4D). The β-ionone ring of the retinal is also shifted towards the cytoplasmic side of the protein by almost 2 Å in the K state. The described structural rearrangements occur synchronously as indicated by the simultaneous appearance of the strong signal at the difference electron density maps near the retinal and W75 and L82 residues. As expected, the RSB is not displaced considerably in the K state. The surrounding of the RSB, such as water molecule w401, residues T74, and S78, as well as the Cl⁻ ion, occupy the same positions as in the ground state of the *Sy*HR photocycle. Thus, the rearrangements of the protein in the K state are concentrated within the retinal-binding pocket; nevertheless, these rearrangements are more pronounced than those observed in proton pumps.

To further characterize the identified 6-s-cis 12-s-cis conformation, we conducted additional MD simulations using the retinal parameters specifically designed to test the torsional energy landscape[35,36]. We note, however, that the environment of *Sy*HR, probably heterogeneous both in vivo and *in meso*, and affecting its dynamics, is not fully characterized. Also, some interactions might be not properly modeled by classical force fields (for example, the stacking interaction between R65 and R120 guanidinium moieties, or the interaction of

E190 and E193 side chains, whose carboxyl oxygen atoms are within ~3.1 Å of each other). To compensate for these uncertainties, we applied position restraints to the *Sy*HR backbone atoms. While these restraints might have affected the observed retinal behavior, we believe that the conducted simulations are informative and indicate absence of strongly energetically unfavorable unrealistic conformations and interactions in the crystallographic models. We found that the retinal, chloride ion close to the Schiff base, water molecules and neighboring side chains maintained their positions throughout the simulations, highlighting stability of the identified 6-s-cis 12-s-cis conformation (Supplementary Fig. 7). Compared to the reference simulations of *Sy*HR in the ground state, the retinal was more mobile in the K state, and L82 assumed multiple conformations (Supplementary Table 2). This probably reflects the fact that while the ground state is truly stable, the K state is an energized metastable intermediate state with an expected half-life of microseconds.

It is important to note that we only hypothesize that the resulting retinal configuration is native to the K state of the protein. We cannot exclude that the resulting retinal conformation is an artifact of the cryotrapping procedure, for example, due to reduced protein mobility in the crystal at 100 K. Nevertheless, non-standard retinal configurations were repeatedly found in MRs. For instance, the ability of MRs, including HRs, to maintain the retinal in atypical (11-cis) configuration as a result of the red-light adaptation was described several times in literature[37,38]. Furthermore, 6-s-*cis* configuration of retinal was already observed in microbial rhodopsins[39]. Finally, the 9-cis and 11-cis isomers can exist in various mutants of bacteriorhodopsin during dark adaptation[40]. The retinal configuration of *Sy*HR in the K state should be verified using other experimental methods, for example, FTIR spectroscopy.

### Discussion

The high resolution structures presented here allow conclusions on the molecular mechanism of chloride transfer. To present date, three groups of MRs are known to pump anions, namely, the best-characterized classical archaeal HRs, eubacterial Cl⁻ pumps, and cyanobacterial HRs, to which *Sy*HR belongs. While eubacterial chloride pumps differ markedly from the other two groups and most likely function via a distinct mechanism, archaeal and cyanobacterial HRs show notable similarities such as trimeric assembly, similar overall architecture of the protomers, and the organization of the RSB region. Phylogenetically, cyanobacterial anion pumps are quite distant to archaeal HRs. DTE proton pumps of cyanobacteria and DTD proton pumps of archaea are closer to cyanobacterial HRs than archaeal HRs (Fig. 1A), however, despite this fact, the photocycles of cyanobacterial anion pumps and archaeal HRs are very similar and consist of K, L₁, L₂, N and O states[16,41]. The question remains whether these two clades of HRs have similar chloride pumping mechanisms or not. In general, we suggest that the mechanism is similar to that proposed for archaeal HRs, such as *Np*HR[23].

The structural data on the ground, K, and O functional states of the *Sy*HR photocycle presented here allowed us to shed light on the molecular mechanism of chloride pumping by the protein. In general, we conclude that the mechanism is similar to that proposed for archaeal HRs, such as *Np*HR[23].

The retinal-binding pocket and the RSB region are organized similarly in *Sy*HR and *Np*HR[42]. In both cases, the chloride ion is coordinated by two residues, serine and threonine, and one water molecule. Notably, the coordination of the chloride ion and water molecule w401 in *Sy*HR is tighter compared to *Np*HR. Indeed, the distance between chloride and the closest water molecule is 3.0 and 3.3 Å in *Sy*HR and *Np*HR, respectively. Also, in *Np*HR, this water molecule forms a weak 3.5 Å H-bond with the RSB, whereas in *Sy*HR the corresponding distance is 2.6 Å (Supplementary Fig. 8A, B). These tighter hydrogen bonds explain well why *Sy*HR has a much higher affinity to chloride

than *Np*HR and *Hs*HR. Indeed, *Sy*HR has a Kd of 0.112 mM, while *Np*HR only 2 mM and *Hs*HR even 10 mM[17,43,44]. Notably, the presence of L49 in *Sy*HR at the position of S81 in *Np*HR also makes the region between the RSB and R71 less hydrated in cyanobacterial rhodopsin.

Comparing the O structures of *Sy*HR and *Np*HR, similarities are evident. Here we should stress that the structure of the O state of *Np*HR was obtained in ref. 23; unfortunately, the coordinates corresponding to this state are not available in the Protein Data Bank (PDB). However, there is a structure of the anion-free blue form of *Np*HR, that resembles the structure of the O state of *Np*HR[21]. Therefore, in this work we used the structure of the blue form of *Np*HR for the comparison of the structures of the O states of *Sy*HR and *Np*HR.

The main structural difference between the ground and the O states of *Np*HR is the flip of T126. In the O state of *Np*HR, the sidechain of T126 occupies the original space of the chloride-binding site. Also, a similar reorientation of S81 is observed, which corresponds to the L49 flip in *Sy*HR (Supplementary Fig. 8C, D).

Otherwise, the structure of the cytoplasmic and intracellular parts of the proteins are mostly the same for the ion-bound and ion-free forms. The exceptions are amino acids Y124 and G234 in *Np*HR. These residues undergo significant rearrangements during the switch between the ion-bound and ion-free forms. Interestingly, *Sy*HR has a corresponding tyrosine (Y72), which remains in the same place, exactly as T183, which corresponds to G234 in *Np*HR.

The remarkable similarity of the anion-free forms and the O states suggests that these proteins have similar chloride pumping mechanisms, although they are phylogenetically quite distant. The high affinity for chloride and the potential ability to pump chloride against strong concentration gradients makes *Sy*HR a good candidate for being a next-generation optogenetic tool.

Based to the available experimental data and the high-resolution structures of *Sy*HR in different functional states presented here, we propose the following key events to occur during the chloride transport by the protein. First, upon absorption of the light photon, retinal isomerizes from the all-*trans* to the 6-s-cis and 12-s-cis configuration in the K state. This isomerization is unique among the known MRs and suggests a unique primary photochemical reaction in *Sy*HR. We should stress that for the further validation of our hypothesis additional experiments are required.

This initial reconfiguration of the retinal does not affect the organization of the RSB region and Cl⁻ ion bound near the RSB. However, residues W75 and L82 of the retinal-binding pocket are rearranged to avoid steric conflict with the cofactor. We suppose that retinal transition from 6-s-*cis*, 12-s-cis to 13-cis occurs later in photocycle, because at this stage we do not observe any relocation of the RSB yet.

Next, during the transitions of *Sy*HR through the L-like and N-like states, the Cl⁻ ion, originally bound in the RSB region, is released to the cytoplasm with the formation of the O state. Our structural data together with the previous spectroscopy studies of the mutant forms of *Sy*HR suggest that Cl⁻ is released through the region of the H167 residue. A large cavity near H167 might serve as part of the ion-release pathway. The mutation of the corresponding histidine to alanine in homologous *Mr*HR results in a dramatic deceleration of the L and N states decay, associated with the ion release. The only alternative route of Cl⁻ release passes through the vicinity of D85, which is not conserved within the family. Moreover, substitution of the D85 aspartate in *Mr*HR corresponding to D85 in *Sy*HR does not strongly affect the photocycle of the protein[16].

We hypothesize that Cl⁻ is translocated within the protein upon the K-to-L and the L-to-N state transitions in the same manner as shown for *Np*HR. We cannot exclude that Cl⁻ ion could also be transiently bound near H167. It should be noted, that despite the absence of counterparts of H167 in archaeal HRs, the Cl⁻ release pathway is formed transiently during their photocycle in the same region of the

protein as we suggest for *Sy*HR. Indeed, according to the structure of the N state of *Np*HR, the chloride releasing pathway is formed near the cytoplasmic part of helix F[23]. In *Np*HR, the appearance of the kink in the helix F at the K215 residue creates a water channel leading outside of the protein. K215 of *Np*HR corresponds to L164 in *Sy*HR, which is located close to H167. Therefore, the release channel in *Np*HR is located in nearly the same area as the proposed anion release pathway in *Sy*HR. This observation highlights again the similarity of the anion pumping mechanisms of archaeal and cyanobacterial HRs.

It remains uncertain which steps of the photocycle correspond to the ion release and uptake in cyanobacterial anion pumps, there is no definitive opinion in the literature. It has been hypothesized that Cl⁻ release occurs upon the L-to-N state transition, whereas the ion uptake is associated with the advent of the O state[17]. Alternatively, it has been suggested that ion release and uptake occur with the formation and the decay of the O state, respectively[18]. The structure of the O state of the *Sy*HR photocycle presented here clearly supports the second idea. Our structural data show that in the O state the retinal is isomerized back to the all-trans configuration, but the Cl⁻ ion is not yet bound in the vicinity of the RSB.

The last step of the photocycle is therefore the uptake of the Cl⁻ ion through the extracellular side of the protein to its original binding site near the RSB. We suggest that the uptake occurs through the large polar concavity protruding from the extracellular bulk in the region between helices A, B, G to the R71 residue as it becomes apparent from the structure of the protein. It is somewhat more difficult to specify what amino acids are involved in chloride uptake. Information from site-specific mutagenesis in *Mr*HR suggests that E193 (E192 in *Mr*HR) can play a role in chloride uptake because its mutation to alanine significantly slows down the turnover of the photocycle[16]. Finally, binding of Cl⁻ to its original site in the central part of the protein results in the *Sy*HR returning to its initial ground state (Fig. 5).

The molecular mechanism of the sulfate pumping in the absence of chloride in *Sy*HR so far remained completely unknown. Our structure of the $SO_4^{2-}$-bound form of *Sy*HR reveals salient details on sulfate transport.

The structural data of the $SO_4^{2-}$ bound protein revealed no sulfate inside the rhodopsin in its ground state, despite the crystals being grown using 2 M $(NH_4)_2SO_4$ as the precipitant. If an interior binding site exists, it must have a very high binding constant. Given the size of the active site of *Sy*HR, even this kind of site is unlikely because there is no space for sulfate in the ground state. However, previous studies have identified a red shift in the absorption spectrum of a protein upon the addition of sulfate to an ion-free protein solution[17]. We suggest that the spectral shift is a result of long-range interactions between the central part of the protein near the retinal cofactor and sulfate ion bound at the site at the surface of *Sy*HR. Indeed, in the sulfate-bound state, the absorption peak of *Sy*HR in violet crystals in the absence of Cl⁻ is 556 nm, which corresponds to the sulfate-bound, but not to the anion-free form of the protein in solution[17].

Thus, our results seem to indicate that *Sy*HR does not bind sulfate inside the protein in the ground state. The ion enters the protein during the photocycle after the photon absorption by the retinal cofactor (Supplementary Fig. 9). Unfortunately, our data is insufficient to precisely identify the stage of the photocycle when binding and translocation of sulfate occur. Although the ground state of $SO_4^{2-}$-bound form is closely similar to the O state of Cl⁻bound *Sy*HR, the structures of the protein in the intermediate states could differ between the $SO_4^{2-}$ and Cl⁻ pumping modes of *Sy*HR.

## Selectivity
The high-resolution structure of *Sy*HR and the previously reported structures of the *Mr*HR and its $SO_4^{2-}$-pumping mutant allow us to infer the key determinants of anion selectivity in cyanobacterial HRs.

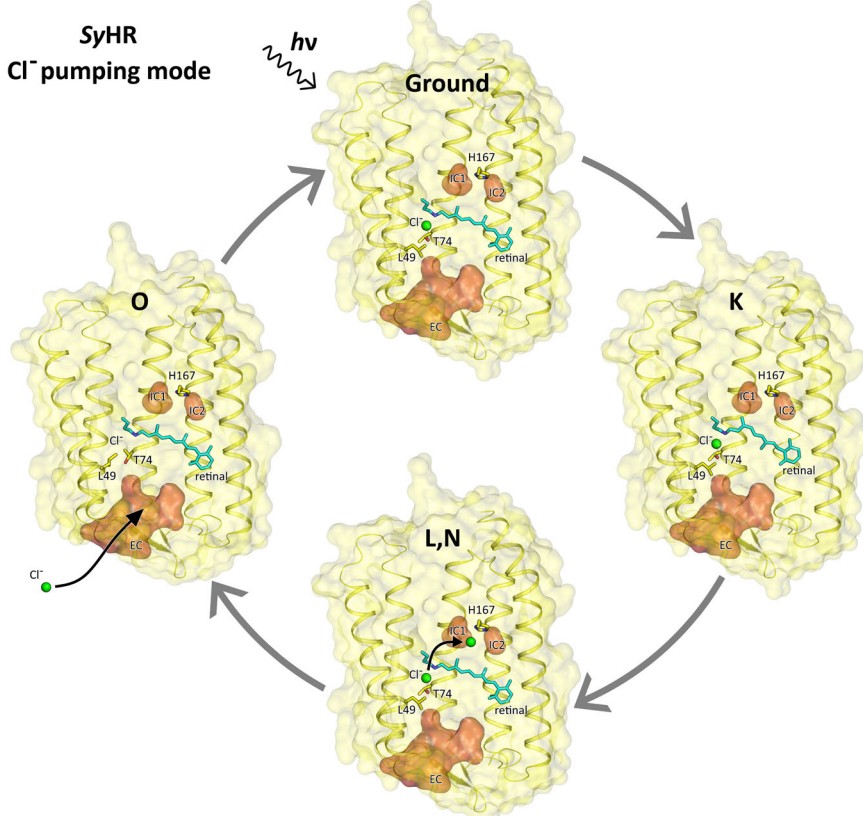

**Fig. 5 | Schematic representation of the chloride pumping mechanism by *Sy*HR during photocycle.** Models of the ground, K, and O states were obtained in the present work. The model of the L,N state is lacking and is shown schematically based on the structure of the ground state of Cl⁻-bound form of *Sy*HR. The side view of *Sy*HR is shown for each state in the way that the cytoplasmic side points to the top and the extracellular side points to the bottom of the figure. Internal cavities were calculated using HOLLOW and are shown as red surface. Cl⁻ ions are shown as green spheres. The retinal cofactor and K205 are colored cyan.

We suggest that the unique ability of cyanobacterial HRs to pump sulfate is due to the presence of a large cavity at the extracellular side of the protein, serving as an ion uptake vestibule. Indeed, in *Sy*HR, the cavity protrudes from the extracellular space through the large pore in the protein to the R71 residue. In the wild-type *Mr*HR, which is incapable of pumping sulfate, the entrance to the pore is much smaller. In *Sy*HR, sulfate likely enters through a large extracellular cavity and then passes through a channel going inside the protein, past the FG loop and more specifically T183. Through this channel, sulfate enters the central part of the protein, where it reaches R71 and T198 and further binds near T74 and S78 in the end of the photocycle. In *Mr*HR, this entrance to the cavity is much tighter and the transition from the cavity to the internal part of the channel is hindered by the presence of bulky E182 instead of T183 in *Sy*HR. Both E182T and E182A variants of *Mr*HR are capable of pumping sulfate, and the E182A mutant pumps sulfate even more efficiently than the *Sy*HR-like E182T mutant[18]. These findings suggest that the size of this amino acid is a determinant of sulfate pumping in cyanobacterial HRs. It should be noted that E182 is conserved in this protein family, in particular, in *Mr*HR, *Np*HR, and *Hs*HR, and is replaced by threonine only in *Sy*HR (Supplementary Fig. 10).

Nevertheless, unlike *Sy*HR, the mutant forms of *Mr*HR seem to have another possible route for the sulfate to reach the internal part of the channel. Thus, double mutation N63A/P118A opens a new path for sulfate located between the BC and DE loops (Supplementary Fig. 11). This double mutant of *Mr*HR pumps sulfate despite the presence of the bulky negatively charged E183 residue. Notably, *Sy*HR also contains alanines in positions corresponding to N63 and P118, but we have not found any additional channel in *Sy*HR in the region between BC and DE

loops. In addition, the entire FG loop of *Sy*HR is substantially different from the corresponding loop of archaeal HRs and also from that of *Mr*HR.

Finally, *Sy*HR and *Mr*HR demonstrate some essential differences between the cytoplasmic internal regions, close to H167, and the putative ion release pathway. Specifically, the cavities in *Sy*HR are notably larger than those in *Mr*HR, which might also be a determinant of the effective translocation of a bulky divalent anion through the cytoplasmic side, which is a specific feature of *Sy*HR.

Taken together, the results of our structural study of *Sy*HR reveal many distinct features of the mechanisms of anion release and uptake in cyanobacterial HRs and suggest potential new optogenetic tools.

## Methods
### Cloning
Nucleotide sequence of *Synechocystis* halorhodopsin (Protein accession no. WP_009632765) was optimized for *E.coli* expression using Thermo Fischer Scientific GeneOptimizer service and synthesized commercially at Eurofins. The optimized gene was introduced into Staby™Codon T7 expression plasmid system (Delphi Genetics, Belgium) via NdeI and XhoI (Thermo Fisher Scientific, USA) that led to the addition of 6×His tag to the C-terminus of the gene. After, selected clones' plasmid DNA was sequenced (Eurofins Genomics, Germany) and *E.coli* C41 strain was transformed with the resulting plasmid DNA construct.

### E. coli expression, solubilization and purification
*E. coli* cells of strain C41 (Lucigen, USA) were transformed with the *Sy*HR expression plasmid. Transformed cells were grown at 37 °C in

shaking baffled flasks in an autoinducing medium ZYP-5052 (36) containing ampicillin (100 mg/liter). When glucose level in the growing bacterial culture dropped below 10 mg/liter, 10 μM all-trans retinal (Sigma-Aldrich, Germany) was added, the incubation temperature was reduced to 20 °C, and incubation continued overnight. Collected cells were disrupted in an M-110P Lab Homogenizer (Microfluidics, USA) at 172.3 MPa in a buffer containing 20 mM Tris-HCl (pH 8.0), 5% glycerol, 0.5% Triton X-100 (Sigma-Aldrich, USA), and deoxyribonuclease I (50 mg/liter) (Sigma-Aldrich, USA). The membrane fraction of cell lysate was isolated by ultracentrifugation at 90,000 g for 1 hour at 4 °C. The pellet was resuspended in a buffer containing 50 mM NaH2PO4/Na2HPO4 (pH 8.0), 0.1 M NaCl, and 1% DDM (Anatrace, Affymetrix, USA) and stirred overnight for solubilization. The insoluble fraction was removed by ultracentrifugation at 90,000 g for 1 hour at 4 °C. The supernatant was loaded on a Ni−nitrilotriacetic acid column (Qiagen, Germany), and SyHR was eluted in a buffer containing 50 mM NaH2PO4/Na2HPO4 (pH 7.5), 0.2 M NaCl, 0.5 M imidazole, and 0.1% DDM. The eluate was subjected to SEC on a 125-ml Superdex 200 prep grade column (GE Healthcare Life Sciences, USA) in a buffer containing 50 mM NaH2PO4/Na2HPO4 (pH 7.5), 0.2 M NaCl, and 0.05% DDM. Protein-containing fractions with the minimal A280/A525 absorbance ratio were pooled and concentrated to 60 mg/ml for crystallization. Cl-free protein was prepared using a similar protocol but without the addition of NaCl and with substitution of tris-HCl (pH 8.0) by HEPES (pH 8.0).

### Crystallization

The crystals were grown with an in meso approach[45], similar to that used in our previous work[46]. The solubilized protein in the crystallization buffer was mixed with pre-melted at 42 °C monoolein (Nu-Chek Prep) to form a lipidic mesophase. The 100 nl aliquots of a protein−mesophase mixture were spotted on a 96-well LCP glass sandwich plate (Marienfeld) and overlaid with 600 nL of precipitant solution using the NT8 crystallization robot (Formulatrix). The best crystals of chloride-bound form (red) were obtained with a protein concentration of 20 mg/ml and the precipitant solution of 1.6 M Ammonium Phosphate pH 4.6. For the chloride-free form (violet) the best crystals were grown with the same protein concentration of 20 mg/ml and the precipitant solution of 2.0 M Ammonium Sulfate, 0.1 M HEPES pH 7.0. The crystals were grown at 20 °C to observable size in two months for the both types. The hexagon-shaped crystals reached 150 μm in length and width with maximum thickness of 10 μm. Crystals of the red type were incubated for 5 min in cryoprotectant solution (1.6 M Ammonium Phosphate pH 4.6, 10% (w/v) glycerol) before harvesting. Crystals of the violet type were incubated for 5 min in cryoprotectant solution (2.0 M Ammonium Sulfate, 0.1 M HEPES pH 7.0, 15% (w/v) glycerol) before harvesting. All crystals were harvested using micromounts (MiTeGen), and were flash-cooled and stored in liquid nitrogen.

### Cryotrapping of the intermediate states of the SyHR photocycle in crystals

Red crystals of SyHR were used for the cryotrapping of active states. For the accumulation and trapping of the K state in the crystals, they were illuminated by the 532-nm laser with the power density of 7.5 mW/cm² for 1 minute at 100 K. The following procedure was performed for the trapping of the O-state. First, the crystal was placed into a cryostream at 100 K. Then, the crystal was illuminated by the 532-nm laser with the power density of 7.5 mW/cm² together with the simultaneous blocking of the cryostream for 2 seconds to allow the crystal to heat up to 293 K. Then the cryostream was released, with the laser remaining switched on to cryotrap the accumulated intermediate. Last, after the release of the cryostream, the laser was switched off.

### Collection and treatment of diffraction data

X-ray diffraction data were collected at PXII beamline of the Swiss Light Source (SLS), Villigen, Switzerland at 100 K, with a PILATUS 6 M detector. We processed diffraction images with XDS[47] and scaled the reflection intensities with AIMLESS from the CCP4 suite[48]. The crystallographic data statistics are presented in Supplementary Table 1. Reference model (ESR, PDB 4HYJ) for molecular replacement was chosen with the RaptorX Structure Prediction Server. Initial phases were successfully obtained in P2₁ space group by the molecular replacement using MOLREP suite. The initial model was iteratively refined using REFMAC5[49], PHENIX and Coot[50,51]. The cavities were calculated using HOLLOW[52]. Hydrophobic-hydrophilic boundaries of the membrane were calculated using PPM server[53].

### Phylogenetic analysis

The microbial rhodopsins data was collected from InterPro[54]. Multiple sequence alignment was performed using Clustal Omega algorithm for generation of phylogenetic tree[55]. The tree was visualized in iTol 6.3.2 and final image was assembled in Adobe Illustrator software[56].

### Molecular dynamics simulations

Cl⁻-bound SyHR in the ground and the K states was simulated in the monomeric form. Hexagonal unit cell contained 47 lipids, 3979 water molecules, 9 Na⁺ and 17 Cl⁻ ions (concentration 0.15 M). SO₄²⁻-bound SyHR in the ground state was simulated in the trimeric form. Hexagonal unit cell contained 88 lipids, 8172 water molecules, 516 NH₄⁺ and 270 SO₄²⁻ ions (concentration 2 M). All atomistic systems were prepared using the Membrane Builder tool[57] of CHARMM-GUI web server[58]. Starting structures included the water molecules, internally bound Cl⁻ ion (in the simulations of monomers), and the interfacial SO₄²⁻ ions (in the simulations of the trimer) observed in the X-ray structures. Positions of other ions were generated using the Monte-Carlo method[59]. Positions of POPC lipids outside the monomer and the trimer were generated automatically. Positions of lipids inside the trimer were generated as follows. We conducted preliminary tests with different numbers of lipids manually placed inside the trimer (5 + 6, 4 + 6, 3 + 6 and 4 + 5 at the cytoplasmic and extracytoplasmic sides, respectively), in accordance with the procedure described in ref. 60, and observed that the 4 + 5 configuration matches the electron densities corresponding to lipids in the best way in 100 ns coarse grained simulation and 20 ns all atom simulation. Coarse grained systems were simulated using Martini 3 force field[61] and converted into all-atom systems using backward[62]. Protonation states of titratable amino acids were assigned using PROPKA3[63,64]; D201 was deprotonated; the Schiff base was protonated.

MD simulations were conducted using GROMACS 2022[65] and the CHARMM36 force field[66,67], TIP3P water model[68], with NBFIX corrections[69] for amine, carboxylate, phosphate and sulfate groups. The parameters for retinal bound to lysine were adapted from ref. 36. Systems were energy minimized using the steepest descent method, thermalized and simulated for 200 ns (for monomers) and 500 ns (for trimer) using the leapfrog integrator with a time step of 2 fs, at a reference temperature of 303.15 K and at a reference pressure of 1 bar. Backbone atoms were harmonically restrained to their experimentally determined positions throughout all of the simulations with a constant of 1000 kJ mol⁻¹ nm⁻². Temperature was coupled using Nosé-Hoover thermostat[70] with coupling constant of 1 ps⁻¹. Pressure was coupled with semiisotropic Parrinello-Rahman barostat[71] with relaxation time of 5 ps and compressibility of 4.510⁻⁵ bar⁻¹.

The simulations were performed using periodic boundary conditions. The center of mass of the reference structure was scaled with the scaling matrix of the pressure coupling. The covalent bonds to hydrogens were constrained using the LINear constraint solver (LINCS) algorithm[72]. The nonbonded pair list was updated every 20 steps with

the cutoff of 1.2 nm. Force based switching function with the switching range of 1.0–1.2 nm and particle mesh Ewald (PME) method with 0.12 nm Fourier grid spacing and 1.2 nm cutoff were used for treatment of the van der Waals and electrostatics interactions. The volumetric density map of $SO_4^{2-}$ ions was obtained using the *VolMap* plugin (version 1.1) for Visual molecular dynamics (VMD)[73] with the sampling density of 1 Å.

## Reporting summary
Further information on research design is available in the Nature Research Reporting Summary linked to this article.

## Data availability
The refined models have been deposited in the Protein Data Bank under PDB identifiers: 7ZOU (Ground Cl⁻), 7ZOV (Ground $SO_4^{2-}$), 7ZOW (K-state), 7ZOY (O-state). Molecular dynamics trajectories have been deposited to Zenodo and are available using the following link: https://doi.org/10.5281/zenodo.6526421. Atomistic model of the starting structure used in molecular dynamics simulations of the sulfate-bound SyHR trimer and the average densities of sulfate sulfur atoms are available as Supplementary Data 1 and 2.

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

## Acknowledgements

We acknowledge the Structural Biology Group of the European Synchrotron Radiation Facility and the European Molecular Biology Laboratory (EMBL) unit in Hamburg at Deutsche Elektronen-Synchrotron (DESY) for granting access to the synchrotron beamlines. Funding: This work was supported by the common program of Agence Nationale de la Recherche (ANR), France and Deutsche Forschungsgemeinschaft, Germany (ANR-15-CE11-0029-02), as well as by funding from Frankfurt: Cluster of Excellence Frankfurt Macromolecular Complexes (to E.B.), by the Max Planck Society (to E.B.) and by the Commissariat à l'Energie Atomique et aux Energies Alternatives (Institut de Biologie Structurale)–Helmholtz-Gemeinschaft Deutscher Forschungszentren (Forschungszentrum Jülich) Special Terms and Conditions 5.1 specific agreement. A.K. was supported by the Ministry of Science and Higher Education of the Russian Federation (agreement 075-03-2022-107, project FSMG-2021-0002). A.A. was supported by the Ministry of Science and Higher Education of the Russian Federation (agreement 075-01645-22-06, project 720000 F.99.1.B385AV67000). Structure analysis was supported by the Russian Science Foundation, grant number 21-64-00018 (to I.G.). This work used the platforms of the Grenoble Instruct Centre (ISBG; UMS 3518 CNRS-CEA-UJF-EMBL) with support from the French Infrastructure for Integrated Structural Biology (ANR-10-INSB-05-02) and GRAL (ANR-10-LABX-49-01) within the Grenoble Partnership for Structural Biology.

## Author contributions

S.V. with the help of T.B. made cloning, expression, and purification of the protein; R.A. performed crystallization of the protein; K.K. and A.R. measured the spectrum in crystals; K.K. and R.A. collected the diffraction data with the help of G.B.; K.K. and S.B. solved the structure; A.K. and I.G. performed molecular dynamics simulations; I.G. helped with structure solving, refinement and analysis; A.A. performed phylogenetic analysis; D.Z. helped with protein expression; D.V. helped with spectrum analysis; V.G. supervised the project; R.A. and S.B prepared pictures; R.A. and V.G. analyzed the results and prepared the manuscript with consultation of G.B, M.E., E.K. and E.B.

## Competing interests

The authors declare no competing interests.
