## [Peer Review File · Nature Communications]

Structural insights into light-driven anion pumping in cyanobacteriaReviewers' Comments:

Reviewer #1:

Remarks to the Author:

The manuscript by Astashkin et al. reports high-resolution crystal structures of anion-pumping rhodopsin (halorhodopsin) from the cyanobacterium *Synechocystis* (SyHR). Halorhodopsins are widely used as optogenetic tools to inhibit neuronal spiking, and the structural information is essential for elucidation of their functional mechanisms and their optimization by molecular engineering. SyHR is the only so far known rhodopsin proposed to pump sulfate besides chloride.

This manuscript is a fundamental study on this protein, and it expands our understanding of all halorhodopsins and even all retinylidene proteins in general. I find particularly interesting the finding of the very unusual 6-*s-cis*, 12-*s-cis* form of the retinal chromophore in the K intermediate of SyHR, although the Authors cannot exclude that it is an artifact of the cryotrapping procedure.

I recommend this manuscript for publication in *Nature Communications* after a minor revision addressing the issues listed below. My major concern is that throughout the manuscript the Authors keep insisting that SyHR pumps chloride and sulfate, whereas in fact these functions, to the best of my knowledge, have never been demonstrated directly, but only inferred from the results of a very indirect assay (Ref. 14). The Authors should be well aware that using such assays have led to a very wrong conclusion at the early stages of research on halorhodopsin. Therefore, I strongly recommend to describe the results obtained with this assay more accurately, as I suggest below. My other concerns are relatively minor.

Lines 26-28: "The only characterized MR that pumps divalent ions belongs to cyanobacterium *Synechocystis* sp. PCC 7509 and is named *Synechocystis* halorhodopsin (SyHR)."

This sentence is misleading, as the reader may assume that "divalent ions" are Ca²⁺. Also, from this sentence it is not clear that SyHR does pump halides, not only "divalent ions". Furthermore, to the best of my knowledge, the only divalent anion tested was sulfate (Ref. 14). Finally, again to the best of my knowledge, the proposed pumping of sulfate by SyHR or MrHR mutants has never been demonstrated by direct photocurrent recording, only by indirect measuring of pH changes in bacterial suspensions (Refs. 14 and 15). Taking all this into account, I suggest to change this sentence to: "The only characterized MR proposed to pump sulfate in addition to halides belongs to the cyanobacterium *Synechocystis* sp. PCC 7509 and is named *Synechocystis* halorhodopsin (SyHR)."

Line 30: "...the structure of its anion-free form..."

The Authors do not report the structure of an anion-free form of SyHR; they do report the structure of its sulfate-bound form (Fig. 3).

Lines 42-43: "...a sequence of structural as well as spectral rearrangements in the protein..."

What are "spectral rearrangements of the protein"? The Authors obviously mean changes in the absorption spectrum during the photocycle, but this has to be stated appropriately.

Lines 46-47: "...as well as sensory and various enzymatic activities..."

To the best of my knowledge, there has been no report yet on any enzymatic activity by any microbial rhodopsin. According to a hypothesis put forward in Ref. 1 cited by the Authors, some channelrhodopsins may activate Ca²⁺ channels by initiating enzymatic cascades, but this hypothesis does not propose that rhodopsin itself acts as an enzyme. There is a large family of multidomain proteins known as "enzymorhodopsins" (for review see [PMID: 30954887]). In this case, photoactivation of the rhodopsin domain results in activation of the enzymatic domain, but again, no

enzymatic function has been attributed to the rhodopsin domain itself.

Lines 56-57: "...is the 3-letter motif TSA instead of the DTD motif that is characteristic of light-driven proton pump bacteriorhodopsin (BR)7."

The reader would be wondering, why are these three residues so important that microbial rhodopsins are classified according to them? Please briefly describe the function of each of these residues in bacteriorhodopsin.

Lines 70-71: "...SyHR has a unique ability to also pump divalent ions, such as SO₄²⁻..."

As mentioned above, no sulfate pumping by SyHR has been shown directly. Please describe the earlier obtained data more accurately, e.g. "Measurements of pH changes in suspensions of E. coli cells that express SyHR have suggested that this protein may also pump sulfate in addition to halides". Also, please replace "divalent ions" with "divalent anions" to avoid confusion.

Line 92: "... (Fig. 1A)18."

Fig. 1B is never mentioned in the manuscript.

Line 159: "... (Fig. 2, D)."

There is no panel D in Figure 2.

Lines 160-161: "The size of IC1 in SyHR is dictated by the conformation of the L170 side chain, which adopts two alternative orientations."

Please indicate the occupancies of these two orientations and indicate that Figure 2B shows the conformation with 3 water molecules in IC1. Is there any functional significance of the existence of these two orientations of the L170 side chain?

Line 178: "...Last frame represents SyHR tetramer..."

Figure 2A clearly shows that the structure is a trimer.

Line 190: "... (Fig. 2, E)."

There is no panel E in Figure 2.

Line 198: "... (Fig. 2, C)."

There is no panel C in Figure 2.

Line 208: "High SO₄²⁻-pumping efficiency was demonstrated for several MrHR mutants..."

As explained above, no direct sulfate pumping has been measured in the cited study (Ref. 15). Please describe it more accurately.

Lines 225-227: "Since SyHR has low K_d to Cl⁻ (0.112 mM), even after using Cl⁻-free solutions during protein purification and crystallization, we found that approx. 25.5% of protein molecules within the violet crystal contain Cl⁻ ion bound near the RSB."

I fail to see the logic here. A low K_d means weak binding, so if the K_d is low, one would expect no Cl⁻ to remain bound to the protein in Cl⁻-free solutions. Do the Authors actually want to emphasize that

the Kd of SyHR is lower than that of HsHR and NpHR (lines 416-417)?

Line 264: "...Last frame represents SyHRv tetramer..."

Figure 3A clearly shows that the structure is a trimer.

Line 300: "...(Fig. 4, D)."

Fig. 4D is mentioned in the text before Fig. 4A, and Fig. 4C is not mentioned in the text at all.

Line 378: "...we only assume that the resulting retinal configuration is native..."

"Hypothesize" seems to be a more appropriate word here than "assume".

Line 403: Delete "we propose" as redundant.

Lines 437-438: "...the ability to pump chloride against strong concentration gradients..."

Again, no chloride pumping ability has been directly demonstrated in SyHR yet.

Line 449: Delete the word "occurs" at the end of this line as duplication.

Line 475: Please indicate in the Fig. 5 caption (or the figure itself) that the molecule is depicted with the extracellular surface pointing downward.

Line 826: I have not found in the manuscript any reference to the PDB accession codes for the four SyHR structures that the Authors have obtained. I assume this is because the manuscript is not yet published, but would like to remind the Authors that this information is required for publication.

Reviewer #2:

Remarks to the Author:

The manuscript by Prof. V. Gordeliy and co-workers focuses on the anion pumping mechanisms observed in the cyanobacterium *Synechocystis halorhodopsin* (SyHR). Studies have suggested that SyHR is the only rhodopsin able to transport sulfate and chloride ions; therefore structural data on such transport mechanisms is very important. In support to their work, the authors present four structures; (1) the crystal structure of the ground state SyHR bound to Cl⁻, (2) the crystal structure of the ground state SyHR bound to SO₄, (3) the crystal structure of the photoreaction intermediate K in the presence of Cl⁻ and (4) the crystal structure of the photoreaction intermediate O in the presence of Cl⁻.

Overall, this is very exciting work that definitely would make an important contribution to the rhodopsin's field. However, I was disappointed to note that although the manuscript describes high quality crystallographic data, its findings weren't well supported by other experimental data including molecular dynamics simulations. For example, the unusual 6-s-cis and 12-s-cis retinal configuration could be supported (or not) by MD simulations. How do we know this unusual retinal configuration isn't result of some crystallographic artefact?...the same goes for the SO₄ binding.

Therefore, I recommend the manuscript publication only after its findings be well supported by additional experimental data.

In addition, a few other notes:

Line 52 The word "was" on "...and was named NpHR" isn't necessary.

Line 140 In figure 1 (B) is difficult to "see" a trimeric perspective. I suggest to add a top view to the figure.

Line 159 Fig.2, D doesn't exist

Line 190 Fig.2, E doesn't exist

Line 198 Fig.2, C doesn't exist

Line 176 In figure 2 B, compartment of the D85 region, the K205 seems to have the same colour as the Retinal in other pictures. It creates confusion as it is also labelled as retinal

Line 222 ".....we assume that the determined structure represents SyHR in the the SO42- -bound form" This is a very "dangerous" sentence. One shouldn't assume anything until solid scientific evidence (at least in written). More peculiar, the authors assume is SO4 -bound form even when experimental data (supplementary Fig 2, A) supports the presence of Cl- ion. No picture showing the 2Fo-Fc electron density around the SO4 is provided, therefore impossible to make comments on the real validation or scattering of the ion.

Line 261/262 Again, Figure 3 panel C seems to suggest that SO42- ion is having a more active role as crystal contact than a physiological one.

Not sure why the sulfate ion valence is difference (from 2- to -1) for the second ion position in panel A and C

Reviewer #3:

Remarks to the Author:

The authors present the crystal structure of SyHR in the ground state, the structure of its anion-free form as well as two photoreaction intermediates, the K and O states. The paper starts with a long introduction, not very easy to follow, for a broad reader, I must say. This introduction presents lots of information and makes it difficult to understand the focus of the article. I suggest simplifying the introduction in order to better highlight the finding of the paper.

In both abstract and introduction the authors stress out the fact that SyHR has a unique ability to also pump divalent ions, such as SO42-. The author's present high-resolution structures of the native light-driven SO42- pump SyHR. They obtained four high-resolution structures of SyHR: Cl- and SO42--bound forms of the ground state and the K and the O intermediate states of the Cl- - pumping mode of the SyHR photocycle. Then the data sheds light on the chloride pumping mechanism, which is quite similar to that of archaeal HRs. However we do not learn much about how the SyHR is supposedly pumping SO42-.

I have one main criticism with this manuscript. The manuscripts lacks Fo-Fc difference maps for the different structures that are described in order to support the claims presented. Considering that most results are high-resolution X-ray structure, this must be corrected. As an example, the Fo-Fc for SO42--bound must be added in order to highlight the density corresponding to SO42-.

In the paragraph « Structure of the O state of the SyHR photocycle » it is indicated: « Fo-Fc difference electron density maps showed a trace amount of the ground state, which was insufficient for proper fitting of this residual conformation. Therefore, we fitted the crystallographic data exclusively with the O state structure in the final model. ». It is again probably important to illustrate this observation by presenting the difference maps and to illustrate better the differences between the two maps.

In discussion: « The main structural difference between the ground and anion-free forms of NpHR is

the flip of T126. Thus, in the O state of NpHR, the side-chain of T126 occupies the original space of the chloride-binding site. Also, a similar reorientation of S81 is observed, which corresponds to the L49 flip in SyHR (Supp. Fig. 3, C,D). « . Again there is no figure showing the map supporting such observation. I feel like this information is important for the reader.

In a general way, figure legends are very shorts and could better describe the findings. For example, which model is presented in figure 1 B, and each monomer could have a different color.

Finally, a recent paper has described the Cl⁻ pumping mechanism in NmHR (Mous et al., 2022, Science 375, 845-851) it would be good to cite the most recent paper.

We greatly acknowledge the work of the reviewers. Their comments helped us to improve the manuscript. In the revised manuscript all comments are addressed.

The most significant change in the article is the correction of the part about sulfate-bound structure. As suggested by the second Reviewer we performed additional study and analysis of our data for the sulfate-bound structure. We performed molecular dynamics simulations of the SyHR^v trimer to identify sulfate binding sites. Simulations showed that sulfate indeed binds to the protein at the site where we observe the first ion in the crystal structure, but doesn't bind at the site of the second ion. Therefore, we decided to further investigate our crystallographic data. In order to get rid of the crystallographic bias, we built $2Fo-Fc$ composite omit map for this region. For the first sulfate ion, a strong density was observed, which described the contours of the sulfate molecule well (Fig. 3C in the revised manuscript). In contrast, the density around the second sulfate was much weaker and could be also interpreted, e.g., as a glycerol molecule. Given our crystallographic data and modeling results, we decided to remove the second ion from our structure.

We also addressed all other comments. Here we provide point by point replies to the reviewers' comments and point out the corresponding modifications of the manuscript done in accordance with the reviewers' suggestions.

Reviewer 1

1) My major concern is that throughout the manuscript the Authors keep insisting that SyHR pumps chloride and sulfate, whereas in fact these functions, to the best of my knowledge, have never been demonstrated directly, but only inferred from the results of a very indirect assay (Ref. 14). The Authors should be well aware that using such assays have led to a very wrong conclusion at the early stages of research on halorhodopsin. Therefore, I strongly recommend to describe the results obtained with this assay more accurately, as I suggest below.

We thank the Reviewer very much for the comment. We corrected our statements on chloride and sulfate pumping of SyHR in the text according to this and following Reviewer's comments. It is important to mention that despite the absence of direct measurements of photocurrent, there are some strong evidences that these proteins are chloride transporters. One of the evidence is the presence of chloride in the structure and its release in the late stages of the photocycle proven with FTIR spectroscopy (ref. 16) and now with our structural data.

2) Lines 26-28: *“The only characterized MR that pumps divalent ions belongs to cyanobacterium Synechocystis sp. PCC 7509 and is named Synechocystis halorhodopsin (SyHR).”*

This sentence is misleading, as the reader may assume that “divalent ions” are Ca²⁺. Also, from this sentence it is not clear that SyHR does pump halides, not only “divalent ions”. Furthermore, to the best of my knowledge, the only divalent anion tested was sulfate (Ref. 14). Finally, again to the best of my knowledge, the proposed pumping of sulfate by SyHR or MrHR mutants has never been demonstrated by direct photocurrent recording, only by indirect measuring of pH changes in bacterial suspensions (Refs. 14 and 15). Taking all this into account, I suggest to change this sentence to: “The only characterized MR proposed to pump sulfate in addition to halides belongs to the cyanobacterium Synechocystis sp. PCC 7509 and is named Synechocystis halorhodopsin (SyHR).”

We thank the Reviewer for this correction. We changed this sentence as the Reviewer proposed.

3) Line 30: *“...the structure of its anion-free form...”*

The Authors do not report the structure of an anion-free form of SyHR; they do report the structure of its sulfate-bound form (Fig. 3).

We apologize for this mistake and corrected “anion-free form” to “sulfate-bound form”.

4) Lines 42-43: *“...a sequence of structural as well as spectral rearrangements in the protein...”*

What are “spectral rearrangements of the protein”? The Authors obviously mean changes in the absorption spectrum during the photocycle, but this has to be stated appropriately.

We agree that this phrase is misleading. We modified it to “it isomerizes to the 13-*cis* conformation and triggers a sequence of changes in the structure as well as in absorption spectrum of the protein”.

5) Lines 46-47: *“...as well as sensory and various enzymatic activities...”*

To the best of my knowledge, there has been no report yet on any enzymatic activity by any microbial rhodopsin. According to a hypothesis put forward in Ref. 1 cited by the Authors, some channelrhodopsins may activate Ca²⁺ channels by initiating enzymatic cascades, but this hypothesis does not propose that rhodopsin itself acts as an enzyme. There is a large family of multidomain proteins known as “enzymerrhodopsins” (for review see [PMID: 30954887]). In this

case, photoactivation of the rhodopsin domain results in activation of the enzymatic domain, but again, no enzymatic function has been attributed to the rhodopsin domain itself.

In this phrase we meant “enzymehodopsins”. Indeed, the enzymatic activity is performed by soluble domains of these proteins, so we agree with the Reviewer and we changed this phrase and added the recommended citation on “enzymehodopsins”. The corrected sentence is as follows: “In nature, MRs perform diverse functions, such as ion pumping and channeling as well as sensory activity and even control of the enzymatic activity^{1,2}.”

6) Lines 56-57: “...is the 3-letter motif TSA instead of the DTD motif that is characteristic of light-driven proton pump bacteriorhodopsin (BR)⁷.”

The reader would be wondering, why are these three residues so important that microbial rhodopsins are classified according to them? Please briefly describe the function of each of these residues in bacteriorhodopsin.

We thank the Reviewer for this suggestion. We included the following brief description of each of the residues of the BR’s DTD motif in the revised manuscript at lines 57-60.

7) Lines 70-71: “...SyHR has a unique ability to also pump divalent ions, such as SO₄²⁻...”

As mentioned above, no sulfate pumping by SyHR has been shown directly. Please describe the earlier obtained data more accurately, e.g. “Measurements of pH changes in suspensions of E. coli cells that express SyHR have suggested that this protein may also pump sulfate in addition to halides”. Also, please replace “divalent ions” with “divalent anions” to avoid confusion.

We thank the Reviewer for this advice. We modified the sentence as Reviewer proposed.

8) Line 92: “....(Fig. 1A)¹⁸.”

Fig. 1B is never mentioned in the manuscript.

We corrected this. Now the reference for Fig. 1B is present (line 129).

9) Line 159: “... (Fig. 2, D).”

There is no panel D in Figure 2.

We apologize for the outdated references. We corrected it to “Fig. 2B”

10) Lines 160-161: *“The size of IC1 in SyHR is dictated by the conformation of the L170 side chain, which adopts two alternative orientations.”*

Please indicate the occupancies of these two orientations and indicate that Figure 2B shows the conformation with 3 water molecules in IC1. Is there any functional significance of the existence of these two orientations of the L170 side chain?

Following the suggestion of the reviewer we specified the occupancies equal to 0.5 for the both conformations and indicated that only one conformation is shown in Fig. 2B (lines 169 and 171). Regarding the functional significance of the two orientations of L170, we cannot exclude that it could have a functional importance as soon as L170 is located in the putative ion release site. But this hypothesis should be tested additionally and is beyond the scope of our manuscript.

11) Line 178: *“...Last frame represents SyHR tetramer...”*

Figure 2A clearly shows that the structure is a trimer.

We apologize for the misprint. We corrected this error.

12) Line 190: *“... (Fig. 2, E).”*

There is no panel E in Figure 2.

We apologize for the outdated references. We changed it to “Fig. 2B”

13) Line 198: *“...(Fig.2, C).”*

There is no panel C in Figure 2.

We apologize for the outdated references. We changed it to “Fig. 2B”

14) Line 208: *“High SO_4^{2-} -pumping efficiency was demonstrated for several MrHR mutants...”*

As explained above, no direct sulfate pumping has been measured in the cited study (Ref. 15). Please describe it more accurately.

We thank Reviewer for the advice. We changed the sentence to the more accurate one: “Experiments showed evidence of SO_4^{2-} -pumping for several *MrHR* mutants, such as E182A, N63A/P118A, and N63A/P118A/E182A.”

15) Lines 225-227: “*Since SyHR has low K_d to Cl^- (0.112 mM), even after using Cl^- -free solutions during protein purification and crystallization, we found that approx. 25.5% of protein molecules within the violet crystal contain Cl^- ion bound near the RSB.*”

I fail to see the logic here. A low K_d means weak binding, so if the K_d is low, one would expect no Cl^- to remain bound to the protein in Cl^- -free solutions. Do the Authors actually want to emphasize that the K_d of SyHR is lower than that of HsHR and NpHR (lines 416-417)?

We thank the Reviewer for this question. In fact, lower dissociation constant (K_d) means a more tightly bound ligand. SyHR has the lowest K_d and thus the highest measured affinity to chloride among all known microbial rhodopsins, which means the strongest binding of chloride to the protein. We suggest this is the reason for the presence of residual chloride even in the structures obtain at Cl^- -free conditions.

16) Line 264: “*...Last frame represents SyHRv tetramer...*”

Figure 3A clearly shows that the structure is a trimer.

We apologize for that misprint. We corrected this error.

17) Line 300: “*...(Fig. 4, D).*”

Fig. 4D is mentioned in the text before Fig. 4A, and Fig. 4C is not mentioned in the text at all.

We thank the Reviewer for this comment. We have swapped panels A,B and C,D to avoid this inconvenience. Also, we added a reference to the ex-panel C (panel A now) on line 308.

18) Line 378: “*...we only assume that the resulting retinal configuration is native...*”

“Hypothesize” seems to be a more appropriate word here than “assume”.

We agree with the reviewer that “hypothesize” is more appropriate word for this sentence. We made the corresponding correction.

19) Line 403: Delete “we propose” as redundant.

We corrected the misspelling.

20) Lines 437-438: “...the ability to pump chloride against strong concentration gradients...”

Again, no chloride pumping ability has been directly demonstrated in SyHR yet.

We thank the reviewer for this remark. We added a word “potential” to this sentence and changed this phrase to “...the potential ability to pump chloride against strong concentration gradients...”

21) Line 449: Delete the word “occurs” at the end of this line as duplication.

We thank the reviewer for this comment and we corrected the mistake.

22) Line 475: Please indicate in the Fig. 5 caption (or the figure itself) that the molecule is depicted with the extracellular surface pointing downward.

We agree with the reviewer that a proper Fig. 5 legend was missing in the original version of the manuscript. We included a detailed Fig. 5 caption in the revised manuscript.

23) Line 826: I have not found in the manuscript any reference to the PDB accession codes for the four SyHR structures that the Authors have obtained. I assume this is because the manuscript is not yet published, but would like to remind the Authors that this information is required for publication.

We thank the reviewer for this reminding. Indeed, we have not fixed the PDB codes in the original manuscript because the manuscript is only under consideration/review while normally we provide PDB codes at a later stage. We agree that it is essential to provide the PDB codes in the revised version of the manuscript, therefore we included the corresponding information to Supplementary Table 1.

Reviewer 2

1) I was disappointed to note that although the manuscript describes high quality crystallographic data, its findings weren't well supported by other experimental data including

molecular dynamics simulations. For example, the unusual 6-s-cis and 12-s-cis retinal configuration could be supported (or not) by MD simulations. How do we know this unusual retinal configuration isn't result of some crystallographic artefact?...the same goes for the SO4 binding.

We thank the reviewer very much for this comment and agree that MD can be used to address the problem. We performed MD simulation to corroborate the crystallographic data and added the results to the manuscript.

Regarding the 6-s-cis 12-s-cis retinal configuration in the K state, we performed the simulations of the ground state as a reference, and of this intermediate state. We used the force field parameters that were based on advanced quantum mechanical calculations to correctly represent the torsional energy landscape of the retinal. Consequently, these parameters are well suited to study possible isomerization of the retinal or to check whether the retinal remains in a stable or metastable conformation. We found that the Schiff base pocket remained stable in both of the simulations (of the ground state and of the K state), and the retinal didn't change its conformation. Thus, MD simulations corroborate the feasibility of the 6-s-cis 12-s-cis retinal configuration.

Regarding the sulfate binding, we performed molecular modelling of SyHR^v in the trimeric form and observed that the sulfate ions bind almost exclusively in the site at the interface between the protomers, where our first sulfate ion was located in the crystal structure (Extended data Fig.4A,B in the revised manuscript). However, we didn't observe noticeable sulfate binding at the second site (on the outward-facing surface of the trimer). Therefore, we decided to check whether we made a mistake by adding a second sulfate to the crystallographic model. In order to get rid of the crystallographic bias, we built *2Fo-Fc* composite omit map for this region. For the first sulfate ion, a strong density was observed, which described the contours of the sulfate molecule well (Fig. 3C in the revised manuscript). In contrast, the density around the second sulfate was much weaker and could be also interpreted, e.g., as a glycerol molecule (see Pic.1 below). Thus, currently, our data fully supports binding of sulfate to the protein at the first site. However, there is not enough support for the second sulfate site, so we decided to remove the corresponding sulfate from the crystallographic model.

Pic. 1 | *2Fo-Fc* composite omit map is countered around the sulfate ions at 1sigma level. For the first ion (right), the strong density is observed, which outlines the molecule. The density around the second ion (left) is much weaker and doesn't outline the molecule.

MD simulations are described on lines 398-401 for the K-state and on lines 249-255 for sulfate binding. MD simulation results are illustrated on Extended Data Fig.4 in the manuscript.

2) Line 52 The word “was” on .”...and was named NpHR” isn't necessary.

We thank the reviewer for this correction. We removed “was” from this sentence.

3) Line 140 In figure 1 (B) is difficult to “see” a trimeric perspective. I suggest to add a top view to the figure.

We changed colors of the different protomers to make it easier to see a trimeric perspective.

4) Line 159 Fig.2, D doesn't exist

Line 190 Fig.2, E doesn't exist

Line 198 Fig.2, C doesn't exist

We apologize for using outdated references. In the revised manuscript all these pictures are in Fig. 2B. We corrected the mistake.

5) Line 176 *In figure 2 B, compartment of the D85 region, the K205 seems to have the same colour as the Retinal in other pictures. It creates confusion as it is also labelled as retinal*

We thank the Reviewer for the correction. Indeed, the K205 was incorrectly signed as retinal in Figures 2, 3, 4 and Extended Data Figures 2, 5, 6. We have corrected this in the revised version of the manuscript. We also indicated in the figure legends that the retinal cofactor and K205 are colored cyan in the figures.

6) Line 222 *“.....we assume that the determined structure represents SyHR in the the SO4²⁻-bound form” This is a very “dangerous” sentence. One shouldn’t assume anything until solid scientific evidence (at least in written). More peculiar, the authors assume is SO4²⁻-bound form even when experimental data (supplementary Fig 2, A) supports the presence of Cl⁻ ion. No picture showing the 2F_o-F_c electron density around the SO4 is provided, therefore impossible to make comments on the real validation or scattering of the ion.*

We agree with the Reviewer that in the original version of our manuscript there were not enough experimental evidences of the sulfate binding to the protein. As it was mentioned above, during revision of the manuscript, we performed MD simulations, which showed sulfate binding to the protein almost exclusively at the place where the first sulfate ion is located in the crystal structure (Extended data Fig.4A,B). Also, as proposed by the reviewer, we built and analyzed 2F_o-F_c omit electron density maps (omitting sulfate ions). The maps strongly support the presence of sulfate ion at the same single site to that demonstrated with molecular dynamics simulations. Therefore, we corrected our structural model so the final model includes only one sulfate ion instead of two originally proposed. We demonstrate 2F_o-F_c omit electron density maps (omitting sulfate ion) in Fig. 3C of the revised manuscript.

7) Line 261/262 *Again, Figure 3 panel C seems to suggest that SO4²⁻ ion is having a more active role as crystal contact than a physiological one.*

Not sure why the sulfate ion valence is difference (from 2- to -1) for the second ion position in panel A and C

We performed MD simulation which corroborated sulfate binding between the two SyHR protomers at the place where we observe the first sulfate ion in our original crystal structure. In the initial version of our manuscript, numbers “1” and “2” corresponded to the numeration of

two sulfate ions and not to the ion valence. Since we removed the second sulfate ion from the crystallographic model, we removed these numbers from the Fig.3, which helps to avoid possible confusion of the readers.

Reviewer 3

1) *I have one main criticism with this manuscript. The manuscript lacks Fo-Fc difference maps for the different structures that are described in order to support the claims presented. Considering that most results are high-resolution X-ray structure, this must be corrected. As an example, the Fo-Fc for SO₄²⁻-bound must be added in order to highlight the density corresponding to SO₄²⁻.*

We thank the Reviewer for the suggestion. We agree, that the demonstration of the difference electron density maps is essential for validation of high-resolution X-ray structures. Therefore, we have revised our manuscript in this regard. We should note, that, in accordance to the comments of Reviewer 2, we performed additional MD simulations to verify sulfate binding to the protein. As a result, we removed one of the two initially proposed sulfate ions at the SyHR surface from the final crystallographic model. We demonstrate $2F_o - F_c$ omit electron density maps for the remaining SO₄²⁻ ion in Fig. 3C in the revised manuscript, which, together with the results of molecular dynamics simulations strongly support sulfate binding to the protein. More information regarding this comment of the reviewer is provided in our replies to the next two comments.

2) *In the paragraph « Structure of the O state of the SyHR photocycle » it is indicated: « Fo-Fc difference electron density maps showed a trace amount of the ground state, which was insufficient for proper fitting of this residual conformation. Therefore, we fitted the crystallographic data exclusively with the O state structure in the final model. ». It is again probably important to illustrate this observation by presenting the difference maps and to illustrate better the differences between the two maps.*

As it is mentioned above, we have built and incorporated the $F_o - F_c$ difference electron density maps corresponding to the data obtained with the crystal after the O state cryotrapping (Extended Data Fig. 2D). As we have mentioned in the manuscript, these maps indicate the trace amount of the ground state of SyHR as clearly seen by a positive peak near the ground state positions of Cl⁻ as well as the T74 and L49 residues.

3) *In discussion: « The main structural difference between the ground and anion-free forms of NpHR is the flip of T126. Thus, in the O state of NpHR, the side-chain of T126 occupies the original space of the chloride-binding site. Also, a similar reorientation of S81 is observed, which corresponds to the L49 flip in SyHR (Supp. Fig. 3, C,D). » . Again there is no figure showing the map supporting such observation. I feel like this information is important for the reader.*

First of all, we apologize for a confusion created by several misprints in the original version of our manuscript. Namely, although we have shown in the Supplementary Fig. 4 of the original manuscript the difference electron density maps corresponding to the K and O intermediate states of SyHR, supporting the findings of our work, we have not referred to this Figure in text. Therefore, in the revised version of the manuscript we refer to this Figure (which is now Extended Data Fig. 6 to follow the formatting guide of *Nature Communications*). We have also modified the first paragraph of the “Structure of the O state of the SyHR photocycle” section as follows to highlight that our analysis is based on various types of electron density maps:

“The O state of the SyHR photocycle was solved at 1.6 Å-resolution. Crystallographic data analysis showed that the major fraction of the protein molecules in the crystal are in the O state. $2F_o-F_c$ and difference $F_{oO}-F_{oGr}$ electron density maps clearly show the structural rearrangements in SyHR associated with the O state formation (Extended Data Fig. 2B, Extended Data Fig. 6A). It needs to be noted, that F_o-F_c difference electron density maps showed a trace amount of the ground state, which was insufficient for proper fitting of this residual conformation (Extended Data Fig. 2D). Therefore, we fitted the crystallographic data exclusively with the O state structure in the final model.” (Lines 334-341)

Since we have already modified the “Results” section in the revised manuscript and showed the difference electron density maps supporting our findings, we do not refer to the maps in the Discussion section to avoid unnecessary repetition.

4) *In a general way, figure legends are very shorts and could better describe the findings. For example, which model is presented in figure 1 B, and each monomer could have a different color.*

We thank the reviewer for this comment. We made the figure legends more detailed. Also we used different colors for each of the monomer in Fig. 1B.

5) *Finally, a recent paper has described the Cl⁻ pumping mechanism in NmHR (Mous et al., 2022, Science 375, 845-851) it would be good to cite the most recent paper.*

We agree that this article is useful for the understanding of the molecular mechanism of chloride pumping by *NmHR* and we added the citation to line 102.

Reviewers' Comments:

Reviewer #1:

Remarks to the Author:

The Authors have revised their manuscript according to my suggestions and the suggestions of the other two Reviewers, including the addition of the 2Fo-Fc composite omit map of the sulfate-binding site. I recommend the revised version for publication.

Reviewer #2:

Remarks to the Author:

Dear Professor V. Gordeliy and Co-authors,

Many thanks for addressing all my previous comments and concerns.

I welcome in particular the addition of the MD simulations experiments to the manuscript that cross-validate the crystallographic findings. I'm also very pleased to see the removal of the second sulfate from the crystallographic model as its presence is not supported neither by the MD experiments neither by the omit electron density maps.

Finally, many thanks for amending all the typos and figure legends and for following Nature Communications formatting.

The findings reported in the manuscript that are important to scientific community in the field are now well justified. Many thanks for your contribution.

Reviewer #3:

Remarks to the Author:

The authors have addressed all my comments

Reviewer #4:

Remarks to the Author:

Comments on 'Structural insights into light-driven anion pumping in cyanobacteria'

This is a technical review of the manuscript entitled 'Structural insights into light-driven anion pumping in cyanobacteria', submitted to Nature Communications.

I have been asked to specifically review the Molecular Dynamics simulations that were performed in response to the first review, in which I did not take part. I do not have specific expertise in the types of proteins/protein complexes discussed in the manuscript. My comments are aimed at assessing the clarity of the simulations described and the inferences drawn from them.

DESCRIPTION OF METHODS

Regarding clarity of presentation, I have a number of comments on the text in the manuscript that describes the simulations done. Based on these descriptions, I do not have a clear enough idea how to reproduce the results.

A. Simulations for ion positions

In the Methods section, the protocol is sketched for setting up atomistic systems of SyHR in the

ground state. For the SO₄-anion, the trimeric form was chosen. In this form, additional lipids may be/are present inside the trimer, and the number and initial positions of these are determined by running coarse-grained simulations 'in accordance with' the protocol as in Ref. 56. Reading that protocol and the text in the Methods section of the present manuscript, I feel that I do not have enough information to successfully reproduce the set-up, possibly because the protocol in Ref. 56 is in my view not complete enough or does not correspond fully to what the authors of the present manuscript did. A more extensive description (in SI for example) will be helpful.

1. As I understand the protocol as described in Ref. 56, the internal lipids in an ATP synthase rotor ring were found after coarse-graining the protein part with martinize.py and embedding this into a membrane with insane.py. What is unclear to me in the present manuscript is the phrase 'Following preliminary tests with different numbers of lipids inside the trimer ...'.

How was the number of lipids inside the trimer varied? Running a number of insane.py scripts with different settings? Taking out lipids after one insane.py run?

What criteria were used to settle on 9 lipids?

2. The internal lipids were used in the set-up of the atomistic simulations. From the protocol in Ref. 56: 'For atomistic simulations, the systems were reassembled using the internal lipid positions taken from CG assembly simulations (Wassenaar et al., 2014), the experimental c-ring structure, and the external lipid positions generated using CHARMM-GUI (Jo et al., 2007).'

Specifically, what tool of CHARMM-GUI was used?

The internal lipids are known after CG simulation in the CG representation. How are they converted to AA representation, and at what point does the 'reassembly' with the CHARMM-GUI generated positions take place? I guess backward.py may have played a role in the protocol, but it is not described in Ref. 56 nor in the present manuscript.

3. The type of lipids is not specified. In the protocol in Ref. 56, POPC lipids were used. In the present manuscript, in Fig. 1B, I see some lipids, but they are long-tail acids of which it is not entirely clear to me whether these are part of the X-ray structure determination.

4. Initial placement of ions is not explicitly described. I assume that ions that were identified in the X-ray structure (like the crystal water) were present, and additional ions added in the CHARMM-GUI set-up. The former is especially important as the authors concluded to remove a previously assigned SO₄ anion position based on the simulations.

Where were ions inside the protein structure located in the starting structures?
How does CHARMM-GUI place other ions?

B. Simulations for retinal conformations

1. In the simulations that test whether the retinal conformations are reasonable, the authors refer to using dihedral parameters from Refs 34, 35, that are 'specifically designed to test the torsional energy landscape'. The point of Refs 34 and 35 is that the parameters in the dihedrals potentials depend on the protonation state of the Schiff base. The authors do not inform us about the protonation state of the Schiff base used, nor how the protonation state is decided on. The work in Ref. 35 examines the

interplay between retinal conformation and protonation state in rhodopsin and finds a strong correlation between them.

In Extended Data Fig. 4C, I see the Schiff base is protonated in both cases. How was this decided on?

Is this important in the ground state versus K state in SyHR?

INFERENCES MADE FROM SIMULATIONS IN RELATION TO REPORTED DATA AND SIMULATION PROTOCOL

A. Simulations for ion positions

The simulations performed to assess likely binding positions for SO₄ anions are not reported in sufficient detail to be able to independently assess the inferences made by the authors.

1. Extended Data Figure 4 A, B shows the position of only a single SO₄ anion in the interface between two subunits of the trimer. The simulation holds 512 SO₄ anions and in the Methods section the VMD VolMap tool is mentioned. The VolMap can be used to show the probability densities, but results of that analysis are not part of any material. The main text does state that multiple binding and unbinding events were observed at or near the crystallographic site.

Is the number of binding/unbinding events enough to estimate a binding constant?

2. In my view, soaking the system with SO₄ anions would serve the purpose to find out whether multiple binding sites exist and where they are, and in addition, possibly find a pathway for their translocation. In this respect, a major worry is the sentence in the Methods section that during the simulations 'Backbone atoms were harmonically restrained to their experimentally determined positions'. It is standard to restrain backbone atoms during minimization and equilibration, but the place of the sentence suggest they were restrained during the 500 ns production simulation. Restraining the backbone positions may block possible pathways and restrict structure remodeling too much.

Were position restraints maintained on the backbone atoms during the production run?

What was the value of the position restraint parameters?

3. On the same theme, to test for a specific possible ion position, I would place it in the proposed place and then observe whether it would like to stay there.

Was the Cl position deeply inside the trimer, near the retinal, tested for its affinity for a SO₄ anion?

4. Combining the themes in points 1 and 3, Extended Fig. 4B, right-hand-side panel, shows 5 snapshots of SO₄ in the binding site. The snapshots are taken 100 ns apart.

Is the lowest position still considered bound?

How is the time progression of the 5 snapshots? Is there a movement out to in or in to out, or have multiple binding/unbinding events taken place? Is it the same SO₄ anion or different ones, exchanged from the solvent?

What happens at the other two sites?

Taken together, I do not see clearly enough how the simulations as described in the revised manuscript contributed to the findings.

By contrast, I find the use of the simulations to review the assignment of the density on the outside of the trimer surface as a possible second binding site valuable and worthy of inclusion in the manuscript or at least the SI. Instead, the presentation ignores this aspect of the process completely, whereas researchers would benefit from learning about this use of simulations.

B. Simulations for retinal position

1. Concerning the simulations for the stability of the position of the retinal in ground and K states, I have the same remark and question regarding the use of restraints of backbone atoms, because restraints during production are likely to lead to artificial stability.

Were position restraints maintained on the backbone atoms during the production run?

I also feel that the authors take minimal information away in the text, just stating that the positions remain stable. In my view, it is clear from the snapshots presented that the variations in the structure are considerably larger in the K state than in the ground state. Given the length of the simulation, these may be larger fluctuations, but, depending on the dihedral potentials used, dihedral transitions are possible on reachable time scales, the larger variation in the K state may point to an oncoming larger structural change.

We greatly acknowledge the work of the Reviewer. The comments helped us to improve the manuscript. In the revised manuscript we described our Molecular Dynamics simulations in more detail and addressed all the Reviewer's comments.

Here we provide point by point replies to the Reviewer's comments and point out the corresponding modifications of the manuscript done in accordance with the Reviewer's suggestions.

Reviewer 4:

Comments on 'Structural insights into light-driven anion pumping in cyanobacteria'

This is a technical review of the manuscript entitled 'Structural insights into light-driven anion pumping in cyanobacteria', submitted to Nature Communications.

I have been asked to specifically review the Molecular Dynamics simulations that were performed in response to the first review, in which I did not take part. I do not have specific expertise in the types of proteins/protein complexes discussed in the manuscript. My comments are aimed at assessing the clarity of the simulations described and the inferences drawn from them.

DESCRIPTION OF METHODS

Regarding clarity of presentation, I have a number of comments on the text in the manuscript that describes the simulations done. Based on these descriptions, I do not have a clear enough idea how to reproduce the results.

Response: *We are grateful to the reviewer for very pointed and extensive comments, and amended the manuscript to include all of the requested details.*

A. Simulations for ion positions

In the Methods section, the protocol is sketched for setting up atomistic systems of SyHR in the ground state. For the SO₄-anion, the trimeric form was chosen. In this form, additional lipids may be/are present inside the trimer, and the number and initial positions of these are determined by running coarse-grained simulations 'in accordance with' the protocol as in Ref. 56. Reading that protocol and the text in the Methods section of the present manuscript, I feel that I do not have enough information to successfully reproduce the set-up, possibly because the protocol in Ref. 56 is in my view not complete enough or does not correspond fully to what the authors of the present manuscript did. A more extensive description (in SI for example) will be helpful.

1. As I understand the protocol as described in Ref. 56, the internal lipids in an ATP synthase rotor ring were found after coarse-graining the protein part with martinize.py and embedding this into a membrane with insane.py. What is unclear to me in the present manuscript is the phrase 'Following preliminary tests with different numbers of lipids inside the trimer ...'.

How was the number of lipids inside the trimer varied? Running a number of insane.py scripts with different settings? Taking out lipids after one insane.py run?

What criteria were used to settle on 9 lipids?

Response: *In the project described in Ref. 56, we simulated capture of lipids inside the ATP synthase rotor ring assembly. We compared the density of captured lipids to the CryoEM density, and found that there were too many lipids. Consequently, we simulated the protein with different numbers of lipids inside the ring and selected the system that matched the experimental densities in the best way.*

In SyHR, the space inside the protein assembly is smaller compared to the space inside the rotor ring of spinach ATP synthase. Consequently, we didn't simulate the assembly process, but placed the lipids manually.

Visual inspection of experimental electron densities indicated that at the cytoplasmic side there are densities corresponding to 9 lipid tail fragments, and seemingly enough free space for at least one more tail fragment; and at the extracytoplasmic side there are also densities corresponding to 9 lipid tail fragments, and seemingly enough free space for at least two more tail fragments. Consequently, initially, we placed 5 and 6 two-tailed lipids at the cytoplasmic and extracytoplasmic sides and conducted CG MD simulations. We found that the overall volume of the lipid patch exceeded the densities observed in the experimental electron densities, and for two of the lipids one of the tails floated on the surface of the lipid patch. Consequently, we've tried systems with 4+6, 3+6 and 4+5 lipids at the cytoplasmic and extracytoplasmic sides, correspondingly, in 100 ns of CG simulation and 20 ns of all atom simulation. We observed that the 4+5 arrangement provides the best agreement with experimental densities, and the tails of the lipids remain buried in the bilayer patch throughout the test simulations.

We have updated the text of the manuscript to include these details (lines 689-700 in the revised manuscript).

2. The internal lipids were used in the set-up of the atomistic simulations. From the protocol in Ref. 56: 'For atomistic simulations, the systems were reassembled using the internal lipid positions taken from CG assembly simulations (Wassenaar et al., 2014), the experimental c-ring structure, and the external lipid positions generated using CHARMM-GUI (Jo et al., 2007).'

Specifically, what tool of CHARMM-GUI was used?

The internal lipids are known after CG simulation in the CG representation. How are they converted to AA representation, and at what point does the 'reassembly' with the CHARMM-GUI generated positions take place? I guess backward.py may have played a role in the protocol, but it is not described in Ref. 56 nor in the present manuscript.

Response: For generating the lipids that are outside of the SyHR trimer, we used CHARMM-GUI Membrane Builder (doi.org/10.1002/jcc.23702). The lipids that we modeled inside the trimer in the CG representation were converted into atomic representation using backward (doi.org/10.1021/ct400617g). We have updated the text of the manuscript to include these details (lines 689-700).

3. The type of lipids is not specified. In the protocol in Ref. 56, POPC lipids were used. In the present manuscript, in Fig. 1B, I see some lipids, but they are long-tail acids of which it is not entirely clear to me whether these are part of the X-ray structure determination.

Response: For MD simulations, we used POPC lipids as these are used commonly in simulations, have average length of hydrocarbon chains and include both saturated and unsaturated chains (16:0-18:1). We have updated the text of the manuscript to include this information (lines 283 and 693).

We also note that Fig. 1B depicts the lipid chains observed in the experimental electron densities. The identity of these chains is not clear as they are partially disordered and head groups usually cannot be identified. The length of the hydrocarbon chains in the lipid fragments included in the crystallographic model does not exceed 18 carbon atoms.

4. Initial placement of ions is not explicitly described. I assume that ions that were identified in the X-ray structure (like the crystal water) were present, and additional ions added in the CHARMM-GUI set-up. The former is especially important as the authors concluded to remove a previously assigned SO₄ anion position based on the simulations.

Where were ions inside the protein structure located in the starting structures?

How does CHARMM-GUI place other ions?

Response: Both sulfate ions that were initially identified in the X-ray structure were present in the starting model for the atomistic MD simulation. No sulfate ions were placed inside the SyHR protomers since none were observed in the X-ray structure. CHARMM-GUI places the solvent ions using the Monte-Carlo ion placing method (doi.org/10.1371/journal.pone.0000880). We have updated the text of the manuscript to include these details (lines 258-265, 668-679).

B. Simulations for retinal conformations

1. In the simulations that test whether the retinal conformations are reasonable, the authors refer to using dihedral parameters from Refs 34, 35, that are ‘specifically designed to test the torsional energy landscape’. The point of Refs 34 and 35 is that the parameters in the dihedrals potentials depend on the protonation state of the Schiff base. The authors do not inform us about the

protonation state of the Schiff base used, nor how the protonation state is decided on. The work in Ref. 35 examines the interplay between retinal conformation and protonation state in rhodopsin and finds a strong correlation between them.

In Extended Data Fig. 4C, I see the Schiff base is protonated in both cases. How was this decided on?

Is this important in the ground state versus K state in SyHR?

Response: *In all simulations reported in this manuscript we modeled the Schiff base in the protonated form. The Schiff base is expected to be in the protonated form in the ground and K states judging from the absorption spectra: microbial rhodopsins with deprotonated Schiff base have an absorption maximum around 400 nm, and SyHR's absorption maxima were above 500 nm. We have updated the text of the manuscript to explicitly describe the protonation state of the Schiff base (lines 680-681).*

The rationale for conducting these MD simulations was as follows: if the reported 6-s-cis, 12-s-cis form of the retinal is strongly energetically unfavorable, we should observe its instability and isomerization in simulations. The parameters reported in Refs 34 and 35 were designed specifically to study retinal isomerization, so we believe that they could be used in our case. We did not observe isomerization of the retinal in our simulations and concluded that the observed conformations are not strongly energetically unfavorable.

INFERENCES MADE FROM SIMULATIONS IN RELATION TO REPORTED DATA AND SIMULATION PROTOCOL

A. Simulations for ion positions

The simulations performed to assess likely binding positions for SO₄ anions are not reported in sufficient detail to be able to independently assess the inferences made by the authors.

1. Extended Data Figure 4 A, B shows the position of only a single SO₄ anion in the interface between two subunits of the trimer. The simulation holds 512 SO₄ anions and in the Methods section the VMD VolMap tool is mentioned. The VolMap can be used to show the probability densities, but results of that analysis are not part of any material. The main text does state that multiple binding and unbinding events were observed at or near the crystallographic site.

Is the number of binding/unbinding events enough to estimate a binding constant?

Response: *Extended Data Figure 4 A shows the probability density of the SO₄ sulfur atoms generated by VolMap with standard settings and contoured at the level of 0.225 a.m.u./Å³. We have added this information to the manuscript. We also have added starting snapshot of the simulation and the calculated density map as a Supplementary Materials 1 and 2.*

We also calculated the distance from each of the identified binding sites to sulfurs of the sulfate ions closest to the site (Extended Data Figure 5A in the revised manuscript). The distributions of these distances show that the binding sites are occupied 49-68% of the time (Extended Data Figure 5B), and the dwell times reach tens of ns (Extended Data Figure 5C). The number of binding events is 592. We have updated the text of the manuscript to include this information (lines 987-996).

2. In my view, soaking the system with SO₄ anions would serve the purpose to find out whether multiple binding sites exist and where they are, and in addition, possibly find a pathway for their translocation. In this respect, a major worry is the sentence in the Methods section that during the simulations ‘Backbone atoms were harmonically restrained to their experimentally determined positions’. It is standard to restrain backbone atoms during minimization and equilibration, but the place of the sentence suggest they were restrained during the 500 ns production simulation. Restraining the backbone positions may block possible pathways and restrict structure remodeling too much.

Were position restraints maintained on the backbone atoms during the production run?

What was the value of the position restraint parameters?

Response: *Yes, the position restraints were maintained during the production run at 1000 kJ mol⁻¹ nm⁻². We have updated the text of the manuscript to include this information (lines 688-689).*

As the goal of our simulation was to probe binding of ions to the structure observed in crystal, we believe that use of the restraints was justified.

Regarding the translocation, we would like to note that SyHR is not a passive but an active transporter; one transport cycle is initiated by absorption of a photon by the retinal, might involve de- and reprotonation events, and takes on average ~25 to ~650 ms to complete. Modeling such processes using all atom molecular dynamics is currently extremely difficult or impossible due to methodological and computational limitations.

Our goal was not to observe possible translocation but rather to check the stability of the ion binding sites on the protein’s surface in this particular chloride-free ground state, and to see whether there are other high-affinity binding sites. The protein’s conformation may change during the photocycle, in which case configuration and affinity of the possible binding sites will change, but accurate description of these events is beyond the scope of the present manuscript.

3. On the same theme, to test for a specific possible ion position, I would place it in the proposed place and then observe whether it would like to stay there.

Was the Cl position deeply inside the trimer, near the retinal, tested for its affinity for a SO₄ anion?

Response: *No, we did not test whether the SO₄ anion will remain bound inside a SyHR protomer at the chloride binding site. Doing a proper simulation would require additional experimental information, which we do not have at the moment.*

SO₄ anion is much larger and more charged compared to Cl anion; if it binds at the same site as Cl anion, the site might be rearranged and/or binding might be transient. Moreover, in the chloride-free structure that we obtained, the side chain of T74 is flipped and occupies the position of the Cl anion.

To test the affinity, we would need to simulate the whole process of ion uptake and/or release, throughout several photocycle intermediates, which at the moment is not feasible, because the expected time scale for a single event (complete photocycle involving binding and release of the anion) is tens to hundreds of milliseconds.

4. Combining the themes in points 1 and 3, Extended Fig. 4B, right-hand-side panel, shows 5 snapshots of SO₄ in the binding site. The snapshots are taken 100 ns apart.

Is the lowest position still considered bound?

How is the time progression of the 5 snapshots? Is there a movement out to in or in to out, or have multiple binding/unbinding events taken place? Is it the same SO₄ anion or different ones, exchanged from the solvent?

What happens at the other two sites?

Response: *We've added extensive quantitative data on the statistics of binding and unbinding of sulfate ions in the Extended Data Fig. 5. There are multiple ions exchanged from the solvent in multiple binding/unbinding events as described in the Extended Data Fig. 5 caption. We also add below a figure showing the time progression of the 5 snapshots for the reviewer's convenience.*

Picture 1. Sulfate-binding site conformations observed in the simulation. 5 snapshots taken at the indicated time stamps are shown.

5. Taken together, I do not see clearly enough how the simulations as described in the revised manuscript contributed to the findings.

By contrast, I find the use of the simulations to review the assignment of the density on the outside of the trimer surface as a possible second binding site valuable and worthy of inclusion in the manuscript or at least the SI. Instead, the presentation ignores this aspect of the process completely, whereas researchers would benefit from learning about this use of simulations.

Response: *We have added a paragraph describing our decision not to assign a sulfate ion to the electron density observed near R146 side chain based on MD simulations (lines 258-265).*

B. Simulations for retinal position

1. Concerning the simulations for the stability of the position of the retinal in ground and K states, I have the same remark and question regarding the use of restraints of backbone atoms, because restraints during production are likely to lead to artificial stability.

Were position restraints maintained on the backbone atoms during the production run?

I also feel that the authors take minimal information away in the text, just stating that the positions remain stable. In my view, it is clear from the snapshots presented that the variations in the structure are considerably larger in the K state than in the ground state. Given the length of the simulation, these may be larger fluctuations, but, depending on the dihedral potentials used, dihedral transitions are possible on reachable time scales, the larger variation in the K state may point to an oncoming larger structural change.

Response: *Yes, position restraints on the backbone atoms were maintained during the production run at 1000 kJ mol⁻¹ nm⁻². We calculated root mean square fluctuations of side chain heavy atom positions in simulations of chloride-bound SyHR in the ground and K states (Supplementary Table 2 in the revised manuscript). Indeed, the retinal is more dynamic in the K state. This probably reflects the fact that while the ground state is truly stable, the K state is an energized metastable intermediate state with an expected half-life at room temperature of microseconds. We have updated the text of the manuscript to include this information (lines 411-414).*

The general rationale for conducting these MD simulations was as follows: if the reported 6-s-cis, 12-s-cis form of the retinal is strongly energetically unfavorable (or absolutely unstable or energetically prohibited), we should observe its instability and isomerization in simulations. We did not observe isomerization of the retinal in our simulations and concluded that the observed conformations are not strongly energetically unfavorable.

*The restraints were added to compensate for the drawbacks of the available approaches. First, this is the lack of the precise knowledge of the environment that the protein finds itself in vivo or in crystals. In both cases the environment is very diverse and heterogeneous; not all lipid species in *Synechocystis* sp. PCC 7509 or in crystals are identified; their enrichment near SyHR has not been determined; interactions with other membrane proteins in the native membrane or crystal contacts in the crystal structure are not modeled explicitly. Besides that, some of the interactions*

present in SyHR are not modeled correctly by the currently available MD force fields. The examples include the stacking interaction between Arg65 and Arg120 guanidinium moieties and the interaction of Glu190 and Glu193 side chains (carboxyl oxygen-oxygen distance of ~ 3.1 Å) near the ion uptake region. Incorrect modeling of these interactions can affect the structural dynamics of the whole protein. Since studying structural dynamics of SyHR by means of molecular dynamics was not the goal of this work, we applied the restraints on the general protein structure and focused on dynamics of the retinal and Schiff base environment.

Reviewers' Comments:

Reviewer #4:

Remarks to the Author:

Review of revised manuscript Structural insights into light-driven anion pumping in cyanobacteria by Astashkin et al.

I thank the authors for responding in detail to my comments and updating the manuscript accordingly. For me, there remains one point of discussion. While I fully understand and acknowledge the limited scope of the MD simulations, I feel that the authors are in danger of overinterpreting their simulation results because the effect of their position restraints on the backbone atoms during the production simulation is unknown and uninvestigated. In the discussion of point B 1, the authors rationalize the application of the restraints by pointing out uncertainties in the experimental conditions in relation to the simulation set-up, as well as the fact that some interactions are not reliably parameterized.

From the response [[my highlighting]]:

'The general rationale for conducting these MD simulations was as follows: if the reported 6-s- cis, 12-s-cis form of the retinal is strongly energetically unfavorable (or absolutely unstable or energetically prohibited), [[we should observe its instability and isomerization in simulations.]] We did not observe isomerization of the retinal in our simulations and concluded that the observed conformations are not strongly energetically unfavorable.'

I could argue that the choices of using the same force constant for all backbone atoms and restraining backbone atoms is arbitrary. I am furthermore not sure that the restraints of the backbone atoms actually allow the retinal to change conformation; the restraints might well be so large that they contribute significantly to the barrier. I therefore feel that the authors should be more careful in their conclusion and add a caveat in the discussion part to remind the reader that position restraints are applied. The paragraph straight after lines 411-414 discusses some other uncertainties regarding this aspect and in my view the addition of a few lines along the lines of the response to add the authors' rationale for applying the restraints is appropriate. Such a caveat could also be helpful to researchers who want to take the simulation data once they are available as a starting point for trying to resolve questions regarding the mechanism.

We are again grateful to the Reviewer for the provided comment and corrected the manuscript accordingly. Please find our response below.

Reviewer 4:

I thank the authors for responding in detail to my comments and updating the manuscript accordingly. For me, there remains one point of discussion. While I fully understand and acknowledge the limited scope of the MD simulations, I feel that the authors are in danger of overinterpreting their simulation results because the effect of their position restraints on the backbone atoms during the production simulation is unknown and uninvestigated. In the discussion of point B 1, the authors rationalize the application of the restraints by pointing out uncertainties in the experimental conditions in relation to the simulation set-up, as well as the fact that some interactions are not reliably parameterized.

From the response [[my highlighting]]:

"The general rationale for conducting these MD simulations was as follows: if the reported 6-s-cis, 12-s-cis form of the retinal is strongly energetically unfavorable (or absolutely unstable or energetically prohibited), [[we should observe its instability and isomerization in simulations.]] We did not observe isomerization of the retinal in our simulations and concluded that the observed conformations are not strongly energetically unfavorable.

I could argue that the choices of using the same force constant for all backbone atoms and restraining backbone atoms is arbitrary. I am furthermore not sure that the restraints of the backbone atoms actually allow the retinal to change conformation; the restraints might well be so large that they contribute significantly to the barrier. I therefore feel that the authors should be more careful in their conclusion and add a caveat in the discussion part to remind the reader that position restraints are applied. The paragraph straight after lines 411-414 discusses some other uncertainties regarding this aspect and in my view the addition of a few lines along the lines of the response to add the authors' rationale for applying the restraints is appropriate. Such a caveat could also be helpful to researchers who want to take the simulation data once they are available as a starting point for trying to resolve questions regarding the mechanism.

Response: *We are grateful to the reviewer for raising this point. Indeed, the applied backbone restraints might contribute to the barrier and energy difference between different retinal conformations. Following the reviewer's suggestion, we extended the corresponding paragraph of the Results section to reflect this fact (lines 370-378).*